

# Seasonal study of stable carbon and nitrogen isotopic composition in fine aerosols at a Central European rural background station

Petr Vodička[1,2], Kimitaka Kawamura[1], Jaroslav Schwarz[2], Bhagawati Kunwar[1], Vladimír Ždímal[2]

[1] Chubu Institute for Advanced Studies, Chubu University, 1200 Matsumoto-cho, Kasugai 487–8501, Japan
[2] Institute of Chemical Process Fundamentals of the Czech Academy of Science, Rozvojová 2/135, 165 02, Prague 6, Czech Republic

*Correspondence to:* vodicka@icpf.cas.cz (P. Vodička), kkawamura@isc.chubu.ac.jp (K. Kawamura)

**Abstract.** Determinations of stable carbon isotope ratios ($\delta^{13}$C) of total carbon (TC) and nitrogen isotope ratios ($\delta^{15}$N) of total nitrogen (TN) were carried out for fine aerosol particles (PM1) collected on a daily basis at a rural background site in Košetice (Central Europe) between 27 September 2013 and 9 August 2014 (n=146). We found a seasonal pattern for both $\delta^{13}$C and $\delta^{15}$N. The seasonal variation in $\delta^{15}$N was more pronounced, with $^{15}$N-depleted values (av. 13.1±4.5‰) in winter and $^{15}$N-enriched values (25.0±1.6‰) in summer. Autumn and spring are transition periods when the isotopic composition gradually changed due to different sources and the ambient temperature. The seasonal variation in $\delta^{13}$C was less pronounced but more depleted in $^{13}$C in summer (-27.8±0.4‰) compared to winter (-26.7±0.5‰).

Major controls of the seasonal dependencies were found based on a comparative analysis with water-soluble ions, organic carbon, elemental carbon, trace gases and meteorological parameters (mainly ambient temperature). A comparison of $\delta^{15}$N with $NO_3^-$, $NH_4^+$ and organic nitrogen (OrgN) revealed that although a higher content of $NO_3^-$ was associated with a decrease in $\delta^{15}$N values in TN, $NH_4^+$ and OrgN had the opposite influences. The highest concentrations of nitrate, mainly represented by $NH_4NO_3$, originated from the emissions from biomass burning, leading to lower $\delta^{15}$N values of approximately 14‰ in winter. During spring, the percentage of $NO_3^-$ in PM1 decreased, and $^{15}$N enrichment was probably driven by equilibrium exchange between the gas and aerosol phases ($NH_3(g) \leftrightarrow NH_4^+(p)$) as supported by the increased ambient temperature. This equilibrium was suppressed in early summer when the $NH_4^+/SO_4^{2-}$ molar ratios reached 2, and nitrate partitioning in aerosol was negligible. During summer, kinetic reactions probably were the primary processes as opposed to gas-aerosol equilibrium on a nitrogen level. However, summertime $\delta^{15}$N values were some of the highest observed, probably suggesting the aging of ammonium sulfate and OrgN aerosols. Such aged aerosols can be coated by organics in which $^{13}$C enrichment takes place by photooxidation process. This result was supported by the positive correlation of $\delta^{13}$C with temperature and ozone, as observed in the summer season.



During winter, we observed an event with the lowest $\delta^{15}N$ and highest $\delta^{13}C$ values. The winter *Event*
was connected with prevailing southeast winds. Although higher $\delta^{13}C$ values probably originated from
biomass burning particles, the lowest $\delta^{15}N$ values were associated with agriculture emissions of $NH_3$
under low temperature conditions that were below 0°C.

## 1. Introduction


Key processes in the atmosphere, which are involved with climate changes, air quality, rain events
(Fuzzi et al., 2015) or visibility (Hyslop, 2009), are strongly influenced by aerosols. Because these
processes are still insufficiently understood, they are studied intensively. One approach to explore
chemical processes taking place in atmospheric aerosols is the application of stable carbon ($\delta^{13}C$) and
nitrogen ($\delta^{15}N$) isotope ratios. These isotopes can provide unique information on source emissions
together with physical and chemical processes in the atmosphere (Gensch et al., 2014; Kawamura et al.,
2004), as well as atmospheric history (Dean et al., 2014). Isotopic composition is affected by both
primary emissions (e.g., Heaton, 1990; Widory, 2006) and secondary processes (e.g., Fisseha et al.,
2009b; Walters et al., 2015a). Both $\delta^{13}C$ and $\delta^{15}N$ values are influenced by kinetic and equilibrium
isotope fractionation that takes place in the atmosphere. In the case of nitrogen, $^{15}N$ is generally depleted
in gas phase precursors (ammonia, nitrogen oxides) but is more enriched in ions ($NH_4^+$, $NO_3^-$) in rainfall
and most enriched in particulate matter and dry deposition (Heaton et al., 1997). In the case of carbon,
the major form is organic carbon (OC), which is composed of large numbers of organic compounds
where isotope fractionations via the kinetic isotope effect (KIE) usually dominating the partitioning
between gas and aerosol (liquid/solid) phases (Gensch et al., 2014).

Many studies have been conducted on $\delta^{13}C$ and $\delta^{15}N$ in particulate matter (PM) in Asia (e.g., Kundu et
al., 2010; Pavuluri et al., 2015b; Pavuluri and Kawamura, 2017) and America (e.g., Martinelli et al.,
2002; Savard et al., 2017). However, only few studies have been performed in Europe. European isotope
studies on aerosols mainly involve the analysis of $\delta^{15}N$ in $NO_3^-$ and/or $NH_4^+$. Widory (2007) published
a broad study on $\delta^{15}N$ in TN in PM10 samples from Paris, focusing on seasons (winter vs. summer)
with some specific sources. Freyer (1991) reported the seasonal variation in $\delta^{15}N$ of nitrate in aerosols
and rainwater as well as gaseous $HNO_3$ at a moderately polluted urban area in Jülich (Germany).
Yeatman et al. (2001a, 2001b) conducted analyses of $\delta^{15}N$ in $NO_3^-$ and $NH_4^+$ at two coastal sites from
Weybourne, England, and Mace Head, Ireland, focusing on the effects of possible sources and aerosol
size segregation on their formation processes and isotopic enrichment. More recently, Ciężka et al.
(2016) reported one-year observations of $\delta^{15}N$ in $NH_4^+$ and ions in precipitation at an urban site in
Wroclaw, Poland, whereas Beyn et al. (2015) reported seasonal changes in $\delta^{15}N$ in $NO_3^-$ in wet and dry
deposition at a coastal and an urban site in Germany to evaluate nitrogen pollution levels.



Studies on δ¹³C at European sites have been focused more on urban aerosols. Fisseha et al. (2009) used
stable carbon isotopes to determine the sources of urban carbonaceous aerosols in Zurich, Switzerland,
during winter and summer. Similarly, Widory et al. (2004) used δ¹³C of TC, together with an analysis
of lead isotopes, to study the origin of aerosol particles in Paris (France). Górka et al. (2014) used δ¹³C
in TC together with PAH analyses for the determination of sources of PM10 organic matter in Wroclaw,
Poland, during vegetative and heating seasons. Masalaite et al. (2015) used an analysis of δ¹³C in TC
on size-segregated urban aerosols to elucidate carbonaceous PM sources in Vilnius, Lithuania. Fewer
studies have been conducted on δ¹³C in aerosols in rural and remote areas of Europe. In the 1990s,
Pichlmayer et al. (1998) conducted an isotope analysis in snow and air samples for the characterization
of pollutants at high-alpine sites in Central Europe. Recently, Martinsson et al. (2017) published
seasonal observations of δ¹³C in TC of PM10 at a rural background station in Vavihill in southern
Sweden based on 25 weekly samples.

These δ¹³C and δ¹⁵N studies show the potential of these isotopes to characterize aerosol types and the
chemical processes that take place in them. To broaden this isotope approach over the European
continent, we present seasonal variations in δ¹³C of total carbon (TC) and δ¹⁵N of total nitrogen (TN)
in the PM1 fraction of atmospheric aerosols at a rural background site in Central Europe. To the best of
our knowledge, this is the first seasonal study of these isotopes in this location, and it is one of the most
comprehensive isotope studies of a fine fraction of aerosols.

## 2.    Materials and methods

### 2.1.    Measurement site


The Košetice observatory is the specialized workplace of the Czech Hydrometeorological Institute
(CHMI), which is focused on monitoring the quality of the environment (Váňa and Dvorská, 2014).
The site is located in the Czech Highlands (49°34'24.13" N, 15°4'49.67" E, 534 m ASL) and is
surrounded by an agricultural landscape and forests, out of range of major sources of pollution with a
very low frequency of traffic. The observatory is officially classified as a Central European rural
background site, which is part of the EMEP, ACTRIS, and GAW networks. A characterization of the
station in terms of the chemical composition of fine aerosols during different seasons and air masses is
presented by Schwarz et al. (2016) and longtime trends by Mbengue et al. (2018) and Pokorná et al.
(2018). As part of a monitoring network operated by the CHMI, the site is equipped with an automated
monitoring system that provides meteorological data (wind speed and direction, relative humidity,
temperature, pressure, and solar radiation) and the concentrations of gaseous pollutants (SO₂, CO, NO,
NO₂, NOₓ and O₃).



## 2.2. Sampling and weighing

Aerosol samples (n = 146) were collected for 24 h every two days from September 27, 2013, to August 9, 2014, using a Leckel sequential sampler SEQ47/50 equipped with a PM1 sampling inlet. Some gaps in sampling were caused by outages and maintenance to the sampler. The sampler was loaded with pre-baked (3 h, 800°C) quartz fiber filters (Tissuequartz, Pall, 47 mm), and the flow rate of 2.3 $m^3$/h was used. In addition, field blanks (n = 7) were also taken for an analysis of the contribution of absorbable organic vapors.

The mass of PM1 was measured by gravimetric analysis of each quartz filter before and after the sampling with a microbalance that had ±1 µg sensitivity (Sartorius M5P, Sartorius AG, Göttingen, Germany). The weighing of samples was performed at 20±1 °C and 50±3 % relative humidity after equilibration for 24 h.

## 2.3. Determination of TC, TN and their stable isotopes

For the TC and TN analyses, small filter discs (area 0.5 $cm^2$, 1.13 $cm^2$ or 2.01 $cm^2$) were placed in a pre-cleaned tin cup, shaped into a small marble using a pair of tweezers, and introduced into the elemental analyzer (EA; Flash 2000, Thermo Fisher Scientific) using an autosampler. Inside the EA, samples were first oxidized in a quartz column heated at 1000°C, in which tin marble burns (~1400°C) and oxidizes all carbon and nitrogen species to $CO_2$ and nitrogen oxides, respectively. In the second quartz column, heated to 750°C, nitrogen oxides were reduced to $N_2$. Evolved $CO_2$ and $N_2$ were subsequently separated on a gas chromatographic column, which was installed in EA, and measured with a thermal conductivity detector for TC and TN. Parts of $CO_2$ and $N_2$ were then transferred into an isotope ratio mass spectrometer (IRMS; Delta V, Thermo Fisher Scientific) through a ConFlo IV interface to monitor the $^{15}N/^{14}N$ and $^{13}C/^{12}C$ ratios.

An acetanilide external standard (from Thermo-electron-corp.) was used to determine the calibration curves before every set of measurements for the calculation of the right values of TC, TN and their isotopic ratios. The $\delta^{15}N$ and $\delta^{13}C$ values of the acetanilide standard were 11.89‰ (relative to the atmospheric nitrogen) and -27.26‰ (relative to Vienna Pee Dee Belemnite standard), respectively. Subsequently, the $\delta^{15}N$ of TN and $\delta^{13}C$ of TC were calculated using the following equations:

$\delta^{15}N$ (‰) = [$(^{15}N/^{14}N)_{sample}$ /$(^{15}N/^{14}N)_{standard}$ − 1]*1000

$\delta^{13}C$ (‰) = [$(^{13}C/^{12}C)_{sample}$ /$(^{13}C/^{12}C)_{standard}$ − 1]*1000



## 2.4. Ion chromatography

Quartz filters were further analyzed by using a Dionex ICS-5000 (Thermo Scientific, USA) ion chromatograph (IC). The samples were extracted using ultrapure water with conductivity below 0.08 µS/m (Ultrapur, Watrex Ltd., Czech Rep.) for 0.5 h using an ultrasonic bath and 1 h using a shaker. The solution was filtered through a Millipore syringe filter with 0.22-µm porosity. The filtered extracts were then analyzed for both anions ($SO_4^{2-}$, $NO_3^-$, $Cl^-$, $NO_2^-$ and oxalate) and cations ($Na^+$, $NH_4^+$, $K^+$, $Ca^{2+}$ and $Mg^{2+}$) in parallel. The anions were analyzed using an anion self-regenerating suppressor (ASRS 300) and an IonPac AS11-HC (2 x 250 mm) analytical column and detected with a Dionex conductivity detector. For cations, a cation self-regenerating suppressor (CSRS ULTRA II) and an IonPac CS18 (2 m x 250 mm) analytical column were used together with a Dionex conductivity detector. The separation of anions was conducted using 25 mM KOH as an eluent at a flow rate of 0.38 ml/min, and the separation of cations was conducted using 25 mM methanesulfonic acid at 0.25 ml/min.

The sum of nitrate and ammonium nitrogen was in good agreement with measured TN (Fig. S1 in Supplementary Information (SI)), and based on the results of TN, $NO_3^-$ and $NH_4^+$, organic nitrogen (OrgN) was also calculated using following equation (Wang et al., 2010): $OrgN = TN – 14*[NO_3^-/62 + NH_4^+/18]$.

## 2.5. EC/OC analysis

Online measurements of organic and elemental carbon (OC and EC) in aerosols were provided in parallel to the aerosol collection on quartz filters mentioned above by a field semionline OC/EC analyzer (Sunset Laboratory Inc., USA) connected to a PM1 inlet. The instrument was equipped with a carbon parallel-plate denuder (Sunset Lab.) to remove volatile organic compounds to avoid a positive bias in the measured OC. Samples were taken at 4 h intervals, including the thermal-optical analysis, which lasts approximately 15 min. The analysis was performed using the shortened EUSAAR2 protocol: step [gas] temperature [°C]/duration [s]: He 200/90, He 300/90, He 450/90, He 650/135, He-Ox. 500/60, He-Ox. 550/60, He-Ox. 700/60, He-Ox. 850/100 (Cavalli et al., 2010). Automatic optical corrections for charring were made during each measurement, and a split point between EC and OC was detected automatically (software: RTCalc526, Sunset Lab.). Instrument blanks were measured once per day at midnight, and they represent only a background instrument signal without any reflection on concentrations. Control calibrations using a sucrose solution were made before each change of the filter (ca. every 2nd week) to check the stability of instruments. The 24 h averages with identical measuring times, such as on quartz filters, were calculated from acquired 4 h data. The sum of EC and OC provided the TC concentrations, which were consistent with TC values measured by EA (see Fig. S2 in SI).





180

### 2.6. Spearman correlation calculations

182

Spearman correlation coefficients (r) were calculated using R statistical software (ver. 3.3.1). Correlations were calculated for the annual dataset (139 samples), separately for each season (autumn - 25, winter - 38, spring - 43, and summer - 33 samples), and the winter *Event* (7 samples). Data from the winter *Event* were excluded from the annual and winter datasets for the correlation analysis. Correlations with p-values over 0.05 were taken as statistically insignificant.

188

### 3. Results and discussion

190

The time series of TN, TC and their isotope ratios ($\delta^{15}N$ and $\delta^{13}C$) for the whole measurement campaign are depicted in Fig. 1. Sampling gaps in autumn and at the end of spring are caused by servicing or outages of the sampler; however, 146 of the samples between September 27, 2013, and August 9, 2014, are enough for a seasonal study. In Fig. 1, the winter *Event* is highlighted, which has divergent values, especially for $\delta^{15}N$, and is discussed in detail in section 3.4.

196

Table 1 summarizes the results for the four seasons: autumn (Sep–Nov), winter (Dec–Feb), spring (Mar–May) and summer (Jun–Aug). The higher TN concentrations were observed in spring (max. 7.59 µgN m$^{-3}$), while the higher TC concentrations were obtained during the winter *Event* (max. 13.6 µgC m$^{-3}$). Conversely, the lowest TN and TC concentrations were observed in summer (Fig. 1).

201

Figure 2 shows relationships between TC and TN and their stable isotopes for one year. The correlation between TC and TN is significant (r=0.70), but during higher concentration events, this dependence can be split due to the different origins of these components. The highest correlations between TC and TN were obtained during transition periods in autumn (0.85) and spring (0.80). Correlations between TC and TN in winter (0.43) and in summer (0.37) were weaker but still statistically significant (p<0.05). As seen in the TC/TN ratios (Table 1), seasonal TC/TN averages fluctuate, but their medians have similar values for autumn, winter and spring, while the summer value is higher (3.45) and roughly points to different aerosol composition in comparison with other seasons. However, seasonal differences between TC/TN ratios are not as large as those in other works (e.g., Agnihotri et al., 2011), and thus, this ratio itself does not provide much information about aerosol sources.


The correlation between $\delta^{13}C$ and $\delta^{15}N$ (Fig. 2, right) is also significant but negative (-0.71). However, there is a statistically significant correlation for spring only (-0.54), while in other seasons, correlations



are statistically insignificant (autumn: -0.29, winter: -0.11 and summer: 0.07). This result shows that
significant and related changes in the isotopic composition of nitrogen together with carbon occur
especially in spring, while there are stable sources of particles during winter and summer. The winter
*Event* measurements show the highest $\delta^{13}C$ values and the lowest $\delta^{15}N$ values, but they are still in line
with the linear fitting of all annual data (Fig. 2, right).

**3.1.**     **Total nitrogen and its $\delta^{15}N$**

The $\delta^{15}N$ values are stable in winter at approximately 15‰, with the exception of the winter *Event*,
which deviated by an average of 13‰. In summer, the $\delta^{15}N$ shows strong enrichment of $^{15}N$ in
comparison with winter, resulting in an average value of 25‰. During the spring period, we observe a
slow increase in $\delta^{15}N$ from April to June (Fig. 1), indicating a gradual change in nitrogen chemistry in
the atmosphere. During autumn, a gradual change is not obvious because of a lack of data in a
continuous time series. Year round, $\delta^{15}N$ ranged from 0.6‰ to 28.2‰. Such a large range may originate
from the limited number of main compounds containing nitrogen in aerosols, which is specifically
present in the form of $NO_3^-$, $NH_4^+$ and/or organic nitrogen (OrgN), and thus, the final $\delta^{15}N$ value in TN
can be formulated by the following equation:
$\delta^{15}N_{TN} = \delta^{15}N_{NO3}*f_{NO3} + \delta^{15}N_{NH4}*f_{NH4} + \delta^{15}N_{OrgN}*f_{OrgN}$
where $f_{NO3} + f_{NH4} + f_{OrgN} = 1$ and f represents the fractions of nitrogen from $NO_3^-$, $NH_4^+$ and OrgN in
TN, respectively. The highest portion of nitrogen is contained in $NH_4^+$ (54 % of TN year-round),
followed by OrgN (27 %) and $NO_3^-$ (19 %). While the $NH_4^+$ content in TN is seasonally stable (51-
58 %, Table 1), the $NO_3^-$ content is seasonally dependent – higher in winter, similarly balanced in spring
and autumn, and very low in summer, when the dissociation of $NH_4NO_3$ plays an important role, and
its nitrogen is partitioned from the aerosol phase to the gas phase (Stelson et al., 1979).

The seasonal trend of $\delta^{15}N$ in TN, with the lowest values in winter and highest in summer, has been
observed in other studies from urban Paris (Widory, 2007), rural Brazil (Martinelli et al., 2002), East
Asian Jeju Island (Kundu et al., 2010) and rural Baengnyeong Island (Park et al., 2018) sites in Korea.
However, different seasonal trends of $\delta^{15}N$ in TN in Seoul (Park et al., 2018) show that such seasonal
variation does not always occur.

Figure 3 shows changes in $\delta^{15}N$ values as a function of the main nitrogen components in TN, with
different colors for different days. There are two visible trends for a type of nitrogen. Although $^{15}N$ is
more depleted with increasing contents of $NO_3^-$ in TN, the opposite is true for $NH_4^+$ and OrgN. The
strongest dependence of most of the bulk data is expressed by a strong negative correlation between
$\delta^{15}N$ and the share of $NO_3^-$ in TN (Fig. 3). In all cases, the dependence during the winter *Event* is



completely opposite to the rest of the bulk data (Fig. 3) showing different processes on $\delta^{15}N$ formation,
which is highlighted by a very strong positive correlation between $\delta^{15}N$ and $NO_3^--N/TN$ (0.98). This
point will be discussed in section 3.4.

Considering the individual nitrogen components, several studies (Freyer, 1991; Kundu et al., 2010;
Yeatman et al., 2001b) show seasonal trends of $\delta^{15}N$ of $NO_3^-$, with the lowest $\delta^{15}N$ in summer and the
highest in winter. Savard et al. (2017 and references therein) summarized four possible reasons for this
seasonality of $\delta^{15}N$ in $NO_3^-$, that is, (i) changes in emissions strength, (ii) influence of wind directions
in the relative contributions from sources with different isotopic composition, (iii) the effect of
temperature on isotopic fractionation and (iv) chemical transformations of nitrogen oxides over time
with a lower intensity of sunlight, which can lead to higher $\delta^{15}N$ values of atmospheric nitrate during
winter months, as shown by Walters et al. (2015a). In the case of our data, mixing of all of these factors
probably had an influence on the nitrate isotopic composition during different parts of the year.

Conversely, Kundu et al. (2010) reported higher $\delta^{15}N$ values of $NH_4^+$ in summer than in winter and
generally reported higher $\delta^{15}N$ values in $NH_4^+$ than in $NO_3^-$ except for winter. In sum, the contribution
of $NH_4^+$ to $\delta^{15}N$ overwhelms the contribution of $NO_3^-$ to $\delta^{15}N$. Additionally, TN is composed of $NH_4^+$,
$NO_3^-$ and OrgN. In Fig. 3, we can observe the enrichment of $^{15}N$ in TN in summer when the lowest
$NO_3^-$ contribution occurs. Thus, higher values of $\delta^{15}N$ in TN in summer are mainly caused by $NH_4^+$
originating from $(NH_4)_2SO_4$, OrgN and ammonium salts of organic acids.

Furthermore, in summer, we observed one of the largest enrichments of $^{15}N$ in TN aerosols in
comparison with other studies (Kundu et al., 2010 and references therein), which may be due to several
reasons. First, the works mentioned above mainly studied total suspended particles (TSP) aerosols;
however, we focus on the fine PM1 fraction, which should be more reactive than the coarse fraction
and consequently result in a higher abundance of $^{15}N$ during the gas/particle portioning of $NH_3$ and
$NH_4^+$. Second, the fine aerosol fraction of the Aitken mode persists for a longer period of time in the
atmosphere than the coarse fraction, which is also a factor leading to higher $^{15}N$ enrichment. Indeed,
Mkoma et al. (2014) reported average higher $\delta^{15}N$ in TN in fine aerosols (17.4‰, PM2.5) in comparison
with coarse aerosols (12.1‰, PM10), and Freyer (1991) also reported higher $\delta^{15}N$ in $NO_3^-$ (4.2‰ to
8‰) in fine aerosols (< 3.5 µm) in comparison with the coarse mode (-1.4‰ to 5.5‰). Third, a shorter
sampling time in this work (24 h) leads to the collection of samples with episodic values (see the winter
*Event*) that would be averaged (overlapped) over a longer time resolution (e.g., weekly samples).

Similarly, as in this study, the highest $\delta^{15}N$ values in TN were observed in a few studies from the Indian
region (Aggarwal et al., 2013; Bikkina et al., 2016; Pavuluri et al., 2010) where biomass burning is



common, and ambient temperatures are high. Therefore, in addition to the above reasons, temperature
also plays a significant role in $^{15}N$ enrichment. This point will be discussed in more detail in section 3.3.

Figure 4 shows the $\delta^{15}N$ in TN as a function of $NO_3$. The $\delta^{15}N$ shows a peak at approximately $14\pm1‰$
with increasing nitrate concentrations. Assuming that $NO_3^-$ in the fine aerosol fraction consists
predominantly of $NH_4NO_3$ (Harrison and Pio, 1983), it can be stated that nitrate at the Košetice site is
a source of nitrogen, with $\delta^{15}N$ values at approximately $14‰$, which is similar to the winter values of
$\delta^{15}N$ in $NO_3^-$ in other studies. Specifically, Kundu et al. (2010) reported a winter average value of $\delta^{15}N$
in $NO_3^-$ at $+15.9‰$ from a Pacific marine site at Gosan Island, South Korea, whereas Freyer (1991)
reported $+9.2‰$ in a moderately polluted site from Jülich, Germany. Yeatman et al. (2001) reported
approximately $+9‰$ from a Weybourne coastal site, UK. Park et al. (2018) reported $11.9‰$ in Seoul
and $11.7‰$ from a rural site in Baengnyeong Island, Korea.

Considering the $\delta^{15}N$ of nitrogen oxides, which are common precursors of particulate nitrate, we can
see that the $\delta^{15}N$ of nitrogen oxides generated by coal combustion (Felix et al., 2012; $+6$ to $+13‰$,
Heaton, 1990) or biomass burning ($+14‰$, Felix et al., 2012) are in a same range with our $\delta^{15}N$ during
the period of enhanced concentrations of $NO_3^-$. These $\delta^{15}N$ values of nitrogen oxides are also
significantly higher than those from vehicular exhaust ($-13$ to $-2‰$ Heaton, 1990; $-19$ to $+9‰$ Walters
et al., 2015b) or biogenic soil ($-48$ to $-19‰$, Li and Wang, 2008). Thus, $\delta^{15}N$ values of approximately
$14‰$ (Fig. 4) are probably characteristic of fresh emissions from heating (both coal and biomass
burning) because these values are obtained during the domestic heating season.

The exponential curves in Fig. 4 represent a boundary in which the $\delta^{15}N$ values are migrating as a result
of enrichment or depletion of $^{15}N$, which is associated with removal or loading of $NO_3^-$ in aerosols.
These curves represent two opposite chemical processes, with a match at approximately $14‰$, which
showed a strong logarithmic correlation (r=0.96 during winter *Event,* green line, and -0.81 for the rest
of points, black line, Fig. S3). These results indicate a significant and different mechanism by which
nitrogen isotopic fractionation occurs in aerosols. In both cases, the decrease in nitrate leads to
exponential changes in the enrichment or depletion of $^{15}N$ from a value of approximately $14‰$. In the
case of enrichment, in addition to a higher proportion of $NH_4^+$ than $NO_3^-$, the dissociation process of
$NH_4NO_3$ can cause an increase in $^{15}N$ in TN during a period of higher ambient temperatures, as
hypothesized by Pavuluri et al. (2010).

OrgN has not been widely studied compared to particulate $NO_3^-$ and $NH_4^+$, although it represents a
significant fraction of TN (e.g., Jickells et al., 2013; Neff et al., 2002; Pavuluri et al., 2015). Figure 5
shows the relationship between $\delta^{15}N$ in TN and OrgN. Organic nitrogen consists organic compounds
containing nitrogen in water soluble and insoluble fractions. The majority of samples have a



concentration range of 0.1-0.5 µg m$^{-3}$ (gray highlight in Fig. 5), which can be considered as background
OrgN at the Košetice site. During the domestic heating season with the highest concentrations of NO$_3^-$
and NH$_4^+$, we can observe a significant increase in OrgN with $\delta^{15}$N again at approximately 14‰, which
implies that the isotopic composition of OrgN is determined by the same process during maximal NO$_3^-$
concentrations, that is, emissions from domestic heating. In the case of emissions from combustion,
OrgN originates mainly from biomass burning (Jickells et al., 2013 and references therein), and thus,
elevated concentrations of OrgN (together with high NO$_3^-$ and NH$_4^+$ conc.) may refer to this source. On
the other hand, looking at the trend of OrgN/TN in dependence on $\delta^{15}$N (Fig. 3), it is more similar to
the trend of NH$_4^+$-N/TN than NO$_3^-$-N/TN. Thus, it can be assumed that changes in the $\delta^{15}$N in OrgN in
samples highlighted as a gray area in Fig. 5 are probably driven more by the same changes in NH$_4^+$
particles, and especially in summer with elevated OrgN in TN (Table 1).

### 3.2.    Total carbon and its $\delta^{13}$C


The $\delta^{13}$C of TC ranged between -25.4‰ and -28.9‰ (Fig. 6), which is similar but broader than the
range reported at a rural background site in Vavihill (southern Sweden, range -26.7 to -25.6‰,
Martinsson et al. (2017)), urban Wroclaw (Poland, range -27.6 to -25.3‰, Górka et al. (2014)), and
different sites (urban, coastal, forest) in Lithuania (East Europe, Masalaite et al., 2015, 2017) but similar
to those published by Fisseha et al. (2009) in Zurich. However, our $\delta^{13}$C values are smaller than those
reported for coastal TSP aerosols from Okinawa (East Asia, range -24.2 to -19.5‰, Kunwar et al.
(2016)) or rural Tanzania (Central-East Africa, range -26.1 to -20.6‰ in PM2.5, Mkoma et al. (2014)).
In fact, similar or different $\delta^{13}$C values are widely reported in the northern and southern hemispheres
(Cachier, 1989), which can be explained by different distributions of C3 and C4 plants (Martinelli et
al., 2002), the influence of marine aerosols (Ceburnis et al., 2016), as well as different anthropogenic
sources (e.g., Widory et al., 2004). The $\delta^{13}$C values at the Košetice site fall within the range common
to other European sites. The $\delta^{13}$C values are significantly smaller than those of $\delta^{15}$N due to a higher
number of carbonaceous compounds in the aerosol mixture whose isotope ratio overlaps each other.
However, it is possible to distinguish lower $\delta^{13}$C values in summer (Table 1), which may indicate a
contribution from higher terrestrial plant emissions. Similarly, Martinsson et al. (2017) reported lower
$\delta^{13}$C values in summer in comparison with other seasons, which they explain by high biogenic aerosol
contributions from C3 plants.

A comparison of $\delta^{13}$C with TC in Fig. 6 shows an enhanced enrichment of $^{13}$C at higher TC
concentrations. The lowest $\delta^{13}$C values were observed in field blank samples (mean -29.2‰, n=7),
indicating that the lowest summer values in particulate matter were close to gas phase values. A similar
dependence of $\delta^{13}$C on the TC concentration was observed by Fisseha et al. (2009), whereby winter $^{13}$C





enrichment was associated with WSOC (water soluble organic carbon) that originated mainly from
wood combustion. Similarly, at the Košetice station, different carbonaceous aerosols were observed
during the heating season (Oct.–Apr.) than in summer (Mbengue et al., 2018; Vodička et al., 2015),
whereby winter aerosols were probably affected by not only biomass combustion but also burning of
coal (Schwarz et al., 2016), which can result in higher carbon contents and more $^{13}$C enriched particles
(Widory, 2006). However, relatively low $\delta^{13}$C values in our range (up to -28.9‰) are caused by not
only sources of TC but also a the fact that fine particles are more $^{13}$C depleted in comparison with coarse
particles (e.g., Masalaite et al., 2015; Skipitytė et al., 2016). Furthermore, based on the number of size
distribution measurements at the Košetice site, larger particles were observed in winter in comparison
with summer, even in the fine particle fraction (Zíková and Ždímal, 2013), which can also have an
effect on lower $\delta^{13}$C values in summer.

### 3.3.    Temperature dependence and correlations of $\delta^{15}$N and $\delta^{13}$C with other variables


Tables 2 and 3 show Spearman's correlation coefficients (r) of $\delta^{15}$N and $\delta^{13}$C with different variables
that may reflect some effects on these isotopes. In addition to year-round correlations, correlations for
each season, as well as for the *Event*, are presented separately.

Correlations of $\delta^{15}$N in winter and summer are often opposite (see e.g., for TN -0.40 in winter vs. 0.36
in summer, for $NH_4^+$ -0.42 in winter vs. 0.40 in summer), indicating that changes in aerosol chemistry
at the nitrogen level are different in these seasons. Similarly, the contradictory dependence between
$\delta^{15}$N and TN in summer and winter was observed by Widory (2007) on PM10 samples from Paris and
was connected with secondary processes affecting the nitrogen chemistry that follows two distinct
pathways between $^{15}$N enrichment (summer) and depletion (winter).

From a meteorological point of view, a significant correlation of $\delta^{15}$N with temperature has been
obtained, indicating the influence of temperature on the nitrogen isotopic composition. Dependence of
$\delta^{15}$N in TN on temperature (Fig. 7) is opposite to that observed by Freyer (1991) for $\delta^{15}$N in $NO_3^-$;
however, it is same to that observed by Ciężka et al. (2016) for $\delta^{15}$N in $NH_4^+$ from precipitation. These
authors concluded that the isotope equilibrium exchange between nitrogen oxides and particulate
nitrates is temperature dependent and could lead to more $^{15}$N enriched $NO_3^-$ during the cold season
(Freyer et al., 1993; Savard et al., 2017). Although Savard et al. (2017) reported a similar negative $\delta^{15}$N
in $NH_4^+$ dependence at temperatures in Alberta (Canada), such as for $NO_3^-$, most studies (e.g.,
Kawashima and Kurahashi, 2011; Kundu et al., 2010) reported the opposite temperature dependence
for $\delta^{15}$N in $NH_4^+$ because the $NH_3$ gas concentrations are more abundant during warm weather





conditions, and thus, isotopic equilibrium exchange $NH_3(g) \leftrightarrow NH_4^+(p)$ leading to $^{15}N$ enrichment in
particles is more intensive.
All the considerations mentioned above indicate that a final relationship between $\delta^{15}N$ in TN and
temperature is driven by the prevailing nitrogen species, which is $NH_4^+$ in our case. A similar
dependence was reported by Pavuluri et al. (2010) between temperature and $\delta^{15}N$ in TN in Chennai
(India), where $NH_4^+$ strongly prevailed. They found the best correlation between $\delta^{15}N$ and temperature
during the colder period (range 18.4-24.5°C, avg. 21.2°C); however, during warmer periods, this
dependence was weakened. In our study, we observed the highest correlation of $\delta^{15}N$ with temperature
in autumn (r=0.58, temp. range -1.9 to 13.9°C, avg. 6.6°C), followed by spring (r=0.52, temp. range
1.5-18.7°C, avg. 9.3°C), but there was even a negative but insignificant correlation in summer (r=-0.21,
temp. range: 11.8-25.5°C, avg. 17.7°C). This result indicates that temperature plays an important role
in the enrichment/depletion of $^{15}N$; however, it is not determined by a specific temperature range but
rather the conditions for repeating the process of "evaporation/condensation", as shown by the
comparison with the work of Pavuluri et al. (2010). It is likely that isotopic fractionation caused by the
equilibrium reaction of $NH_3(g) \leftrightarrow NH_4^+(p)$ reaches a certain level of enrichment under higher
temperature conditions in summer.

In summer, $\delta^{15}N$ correlates positively with $NH_4^+$ (r=0.40) and $SO_4^{2-}$ (0.51), indicating a link with
$(NH_4)_2SO_4$ that is enriched by $^{15}N$ due to aging. Figure 8 shows a decreasing molar ratio of $NH_4^+/SO_4^{2-}$
with increasing $^{15}N$ enrichment, especially during spring, indicating a gradual uptake of ammonia in the
gas phase to aerosol phase. With a decreasing $NH_4^+/SO_4^{2-}$ molar ratio, there is also a visible decrease in
the nitrate content in aerosols (Fig. 8). However, when the $NH_4^+/SO_4^{2-}$ ratio approaches a value below
2, there is not enough available ammonia in the gas phase, leading to the exclusion of nitrate from the
aerosol phase, as well as to the disruption of the thermodynamic equilibrium between $NH_3(g) \leftrightarrow$
$NH_4^+(p)$, which previously led to $^{15}N$ enrichment in the particles. In this context, we note that 25 out of
33 summer samples have molar $NH_4^+/SO_4^{2-}$ ratios below 2, and the remaining samples are
approximately 2, although the average relative abundance of $NO_3^-$ in PM1 in those samples is very low
(ca. 1.7 %).

Recently, Silvern et al. (2017) reported that organic aerosols can play a role in modifying or retarding
the achievement of $H_2SO_4$-$NH_3$ thermodynamic equilibrium at $NH_4^+/SO_4^{2-}$ ratios of less than 2, even
when sufficient amounts of ammonia are present in the gas phase. Thus, an interaction between sulfates
and ammonia may be hindered such that organics coated with aged aerosols preferentially react (Liggio
et al., 2011). Indeed, we observed a positive (and significant) correlation between temperature and $\delta^{13}C$
(r=0.39) only in summer, whereas $\delta^{15}N$ vs. temperature is negative (-0.21), suggesting that the
thermodynamic equilibrium between $NH_3$ (g) and nitrogen in particles was minimal or replaced by the
influence of organics in this season. Ammonia measurements directly at the Košetice site were carried





out until 2001, and they showed that the $NH_3$ concentrations in summer and winter were comparable
(http://portal.chmi.cz/files/portal/docs/uoco/isko/tab_roc/2000_enh/CZE/kap_18/kap_18_026.html),
which indirectly support the above hypothesis.

The summer positive correlations of $\delta^{13}C$ with ozone (r=0.66) and temperature (0.39) indicate oxidation
processes that can indirectly lead to carbon isotope enrichment. This result is also supported by the fact
that the content of oxalate in PM1, measured by IC, was twice as high in spring and summer than in
winter and autumn. The influence of temperature on $\delta^{13}C$ in winter is opposite to that in the summer.
The winter negative correlation (-0.35) probably points to the evolution of more fresh emissions from
domestic heating with higher contents of $^{13}C$ during lower temperatures.

The whole year temperature dependence on $\delta^{13}C$ is the opposite of that observed for $\delta^{15}N$ (Fig. 7, left),
suggesting more $^{13}C$-depleted products in summer. This result is probably connected with different
carbonaceous aerosols during winter (anthropogenic emissions from coal, wood and biomass burning
with the enrichment of $^{13}C$) in comparison with the summer season (primary biogenic and secondary
organic aerosols with lower $\delta^{13}C$). The data of $\delta^{13}C$ in Fig. 7 are also more scattered, which indicates
that in the case of carbon, the isotopic composition depends more on sources than on temperature.

Correlations of $\delta^{13}C$ with OC are significant in all seasons; they are strongest in spring and weakest in
summer (Table 3). Correlations of $\delta^{13}C$ with EC, whose main source is combustion processes from
domestic heating and transportation, are significant (r=0.61-0.88) only during the heating season
(autumn–spring, see Table 3), while in summer, the correlation is statistically insignificant (0.28). Thus,
the isotopic composition of aerosol carbon at the Košetice station is not significantly influenced by EC
emitted from transportation, otherwise the year-round correlation between $\delta^{13}C$ and EC would also be
significant in summer. This result is consistent with positive correlations between $\delta^{13}C$ and gaseous
$NO_2$, as well as particulate nitrate, which is also significant from autumn to spring, and this result is
also supported by the negative correlation of $\delta^{13}C$ with the EC/TC ratio (r=-0.51), which is significant
only in summer.

It should be mentioned that the wind directions during the campaign were similar, with the exception
of the winter season, when southeast (SE) winds prevailed (see Fig. S4 in SI). We did not observe any
specific dependence of isotopic values on wind directions, except for the *Event*.





### 3.4. Winter *Event*

The winter *Event* represents a period between January 23 and February 5, 2014, when enrichment of $^{13}C$ and substantial depletion of $^{15}N$ occurred in PM1 (see Figs. 1 and 9 for details). We do not observe any trends of the isotopic compositions of $\delta^{15}N$ and $\delta^{13}C$ with wind directions, except for the period of the *Event* and one single measurement on 18[th] December 2013. Both the *Event* and the single measurement are connected to SE winds through Vienna and the Balkan Peninsula (Fig. 10). More elevated wind speeds with very stable SE winds are observed on the site with samples showing the most $^{15}N$ depleted values at the end of the *Event* (Fig. 9). Stable weather conditions and the homogeneity of the results indicate a local or regional source, which is probably associated with emissions of sulfates (Fig. S5), which are not sufficiently mixed at this time.

Although the *Event* contains only 7 samples, high correlations are obtained for $\delta^{15}N$ and $\delta^{13}C$ (Tables 2 and 3). Generally, correlations of $\delta^{15}N$ with several parameters during the *Event* are opposite to those of the four seasons, indicating the exceptional nature of these aerosols from a chemical point of view. During the *Event*, $\delta^{15}N$ correlates positively with $NO_3^-$ (r=0.96) and $NO_3^-$-N/TN (0.98), with large values of $\delta^{15}N$ at approximately 14‰, which we previously interpreted as the emissions from domestic heating by coal and/or biomass burning. Positive correlations of $\delta^{13}C$ with oxalate and potassium (both 0.93) and the negative correlation with temperature (-0.79) also show that the *Event* is associated with emissions from combustion.

In contrast, we find that most $\delta^{15}N$ values with a depletion of $^{15}N$ are associated with enhanced $NH_4^+$ contents (70-80 %) and the almost total absence of $NO_3^-$ nitrogen (see Figs. 3 and 4). Although some content of OrgN is detected during the *Event* (Fig. 3), the correlation between $\delta^{15}N$ and OrgN/TN is not significant (Table 2). This result shows that nitrogen with the lowest $\delta^{15}N$ values is mainly connected with $NH_4^+$, which is supported by a strong negative correlation between $\delta^{15}N$ and $NH_4^+$/TN (-0.86). Assuming that nitrogen in particles mainly originates from gaseous nitrogen precursors via gas-to-particle conversion (e.g., Wang et al., 2017) during the *Event,* we should expect the nitrogen to originate mainly from $NH_3$ with depleted $^{15}N$ but not nitrogen oxides. Agricultural emissions from both fertilizer application and animal waste are such sources of $NH_3$ emissions (Felix et al., 2013). Considering possible agriculture emission sources, there exist several collective farms, with both livestock (mainly cows, Holsteins cattle) and crop production in the SE direction from the Košetice observatory – namely, Agropodnik Košetice (in 3.4 km distance), Agrodam Hořepník (6.8 km) and Agrosev Červená Řečice (9.5 km). Skipitytė et al. (2016) reported lower $\delta^{15}N$ values of TN (+1 to +6‰) for agriculture-derived particulate matter of poultry farms, which are close to our values obtained during the *Event*.



The $\delta^{15}N$ values from the *Event* are associated with an average temperature of below 0°C (Figs. 7 and
9). Savard et al. (2017) observed the lowest values of $\delta^{15}N$ in $NH_3$ with temperatures below -5°C, and
the $NH_4^+$ particles that were simultaneously sampled were also isotopically lighter compared to the
samples collected under higher temperature conditions. They interpreted this result as the preferential
dry deposition of heavier isotopic $^{15}NH_3$ species during the cold period, whereas the remaining lighter
$^{14}NH_3$ species in the atmosphere, lead to lighter $NH_4^+$ in particles. Moreover, the removal of $NH_3$ by
dry deposition also leads to a non-equilibrium state between the gas and aerosol phases. Such an absence
of equilibrium exchange of $NH_3$ between the gas and liquid/solid phases is supported by a $NH_4^+/SO_4^{2-}$
molar ratio below 2 for the three most $^{15}N$ depleted samples (Fig. 8). In such conditions, nitrate
partitioning in PM is negligible, and unidirectional reactions of lighter $NH_3$ isotope with $H_2SO_4$ in the
atmosphere are strongly preferred due to the kinetic isotope effect, which is (after several minutes)
followed by enrichment of the nitrogen due to the newly established equilibrium (Heaton et al., 1997).
Based on laboratory experiments, Heaton et al. (1997) estimated the isotopic enrichment factor between
gas $NH_3$ and particle $NH_4^+$, $\varepsilon_{NH4-NH3}$, to be +33‰. Savard et al. (2017) reported an isotopic difference
($\Delta\delta^{15}N$) between $NH_3$ (g) and particulate $NH_4^+$ as a function of temperature, whereas $\Delta\delta^{15}N$ for a
temperature of approximately 0°C was approximately 40‰. In both cases, after subtraction of these
values (33 or 40‰) from the $\delta^{15}N$ values of the measured *Event*, we obtain values between
approximately -28 to -40‰, which are in a range of $\delta^{15}N$-$NH_3$ (g) measured for agricultural emissions.
These values are especially in good agreement with $\delta^{15}N$ of $NH_3$ derived from cow waste (ca. -22 to -
38‰, Felix et al., 2013).

Thus, in case of the *Event*, we probably observe PM representing a mixture of aerosols from household
heating characterized by higher amounts of $NO_3^-$ and $\delta^{15}N$ in TN (ca. 14‰), which are gradually
replaced by $^{15}N$-depleted agricultural aerosols. Results of the whole process from low temperatures that
first support dry deposition of $NH_3$ followed by unidirectional (kinetic) reaction of lighter isotope
$NH_3(g) \rightarrow NH_4^+(p)$ originate mainly from agricultural sources in the SE direction from the Košetice
station.

If the four lowest values of $\delta^{15}N$ mainly represent agricultural aerosols, then the $\delta^{13}C$ values from the
same samples should also be characteristic of agricultural sources. In this case, the $\delta^{13}C$ values ranging
from -25.4 to -26.2‰ belong to the most $^{13}C$ enriched fine aerosols at the Košetice site. However,
similar $\delta^{13}C$ values were reported by Widory (2006) for particles from coal combustion. Skipitytė et al.
(2016) reported a mean value of $\delta^{13}C$ in TC (-23.7±1.3‰) for PM1 particles collected on a poultry farm,
and they suggested the litter as a possible source for the particles. Thus, in the case of $\delta^{13}C$ during the
*Event* we observed, emissions either from domestic heating and/or agricultural sources are responsible
for the $^{13}C$ values.



## 4. Summary and Conclusions

Based on the analysis of year-round data of stable carbon and nitrogen isotopes, we extracted some important information on the processes taking place in fine aerosols during different seasons at the Central European station of Košetice. Seasonal variations were observed for $\delta^{13}C$ and $\delta^{15}N$, as well as for TC and TN. The supporting data (i.e., ions, EC/OC, meteorology, trace gases) revealed characteristic processes that led to changes in the isotopic compositions on the site.

The main and gradual changes in nitrogen isotopic composition occurred in spring. During early spring, domestic heating with wood stoves is still common, with high nitrate concentrations in aerosols, which decreased toward the end of spring. Additionally, temperature slowly increases, and the overall situation leads to thermodynamic equilibrium exchange between gas ($NO_{x3}$-$NH_3$-$SO_2$ mixture) and aerosol ($NO_3^-$ - $NH_4^+$- $SO_4^{2-}$ mixture) phases, which causes $^{15}N$ enrichment in aerosols. Enrichment of $^{15}N$ ($\Delta\delta^{15}N$) from the beginning to the end of spring was approximately +10‰. Gradual springtime changes in isotopic composition were also observed for $\delta^{13}C$, but the depletion was small, and $\Delta\delta^{13}C$ was only -1.4‰.

In summer, we observed the lowest concentrations of TC and TN; however, there was an enhanced enrichment of $^{15}N$, which was probably caused by the aging of nitrogen aerosols, where ammonium sulfate is subjected to isotopic fractionation via equilibrium exchange between $NH_3(g)$ and $NH_4^+(p)$. Based on a $NH_4^+$/$SO_4^{2-}$ molar ratio of less than 2, we concluded that summer aerosols become more acidic, and thus, kinetic isotopic fractionation took place via the equilibrium exchange of nitrogen species. However, summer values of $\delta^{15}N$ were still among the highest compared with those in previous studies, which can be explained by several factors. First, a fine aerosol fraction (PM1) is more reactive, and its residence time in the atmosphere is longer than coarse mode, leading to $^{15}N$ enrichment in aged aerosols. Second, summer aerosols, compared to other seasons, contain a negligible amount of nitrate, contributing to a decrease in the average value of $\delta^{15}N$ of TN. On the other hand, we observed an enrichment of $^{13}C$ only in summer, which can be explained by the photooxidation processes of organics and is supported by the positive correlation of $\delta^{13}C$ with temperature and ozone. Despite this slow enrichment process, summertime $\delta^{13}C$ values were the lowest compared to those in other seasons and referred predominantly to organic aerosols of biogenic origin. The role of organics in summer may also have an effect on the aforementioned $^{15}N$ enrichment due to thermodynamic equilibrium.

In winter, we found the highest concentrations of TC and TN. Lower winter $\delta^{15}N$ values were apparently influenced by fresh aerosols from combustion, which were strongly driven by the amount of nitrates (mainly $NH_4NO_3$ in PM1), and led to an average winter value of $\delta^{15}N$ approximately 14‰. Winter $\delta^{13}C$ values were more enriched than summer values, and they were connected mainly to emissions from coal and (mostly) biomass burning for domestic heating.

We observed an aerosol event in winter, which was characterized by temperatures below the freezing point, stable southeast winds, and a unique isotope signature with a depletion of $^{15}N$ and enrichment of



$^{13}$C. The winter *Event* characterized by $^{15}$N depletion was probably caused by the dry deposition of NH$_3$
(with heavier isotope) during cold weather, and with decreasing concentrations of NO$_3^-$. However, it
was completely opposite to a summertime decrease in nitrate, which led to an enrichment of $^{15}$N. In the
case of the most depleted $^{15}$N event, nitrate was suppressed to partition in aerosol and gas phases with
unidirectional reactions of isotopically light ammonia and sulfuric acid resulting in (NH$_4$)$_2$SO$_4$, which
originated mainly from agriculture emissions in this case.
The majority the yearly data showed a strong correlation between $\delta^{15}$N and ambient temperature,
demonstrating an enrichment of $^{15}$N via isotopic equilibrium exchange between the gas and particulate
phases. This process seemed to be one of the main mechanisms for $^{15}$N enrichment at the Košetice site,
especially during spring. The most $^{15}$N-enriched summer and most $^{15}$N-depleted winter samples were
limited by the partitioning of nitrate in aerosols and suppressed equilibrium exchange between gaseous
NH$_3$ and aerosol NH$_4^+$.
This study revealed a picture of the seasonal cycle of $\delta^{15}$N in aerosol TN at the Košetice site. In the case
of carbon, the seasonal cycle of $\delta^{13}$C values was not so pronounced because they mainly depend on the
isotopic composition of primary sources, which often overlapped, and because secondary reactions
were influenced by the kinetic isotopic effect, while phase transfer probably did not play a crucial role.

**Acknowledgements**


This study was supported by funding from the Japan Society for the Promotion of Science (JSPS)
through Grant-in-Aid No. 24221001, from the Ministry of Education, Youth and Sports of the Czech
Republic under the project No. LM2015037 and under the grant ACTRIS-CZ RI
(CZ.02.1.01/0.0/0.0/16_013/0001315). We also thank the Czech Hydrometeorological Institute for
providing its meteorological data and Dr. Milan Váňa and his colleagues from the Košetice Observatory
for their valuable cooperation during the collection of samples. We appreciate the financial support of
the JSPS fellowship to P. Vodička (P16760).

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








**Tables:**


Table 1: Seasonal and entire campaign averages ± standard deviations, (medians in brackets) of
different variables.

| | Autumn | Winter | Spring | Summer | Year |
|---|---|---|---|---|---|
| **N of samples** | 25 | 45 | 43 | 33 | 146 |
| **TC [µg m⁻³] (from EA)** | 3.61±1.61 (3.30) | 4.76±2.44 (3.88) | 3.78±2.03 (3.04) | 2.71±0.76 (2.68) | 3.81±2.03 (3.35) |
| **TN [µg m⁻³]** | 1.56±1.18 (1.33) | 1.67±0.96 (1.45) | 2.00±1.62 (1.47) | 0.81±0.29 (0.82) | 1.56±1.22 (1.26) |
| **$\delta^{13}C$ [‰]** | -26.8±0.5 (-26.9) | -26.7±0.5 (-26.7) | -27.1±0.5 (-27.0) | -27.8±0.4 (-27.7) | -27.1±0.6 (-27.0) |
| **$\delta^{15}N$ [‰]** | 17.1±2.4 (16.9) | 13.1±4.5 (15.2) | 17.6±3.5 (17.3) | 25.0±1.6 (25.1) | 17.8±5.5 (16.9) |
| **TC/PM1 [%]** | 28±6 (26) | 33±8 (32) | 38±15 (35) | 31±6 (30) | 33±11 (31) |
| **TN/PM1 [%]** | 11±3 (11) | 11±3 (12) | 17±4 (17) | 9±2 (9) | 12±4 (12) |
| **$NO_3^-$-N/TN [%]** | 21±6 (21) | 25±8 (28) | 22±8 (21) | 5±3 (4) | 19±10 (20) |
| **$NH_4^+$-N/TN [%]** | 51±6 (51) | 51±9 (49) | 58±7 (60) | 57±6 (57) | 54±8 (54) |
| **OrgN/TN [%]** | 28±8 (26) | 25±8 (23) | 20±8 (19) | 39±6 (38) | 27±10 (25) |
| **TC/TN** | 2.77±1.10 (2.60) | 3.34±1.66 (2.68) | 2.33±0.98 (2.34) | 3.60±1.23 (3.45) | 3.01±1.38 (2.61) |




















Table 2: Spearman correlation coefficients (r) of $\delta^{15}$N with various tracers. Only bold values are statistically significant (p-values < 0.05).

| $\delta^{15}$N vs. | Autumn | Winter* | Spring | Summer | Year* | Event |
|---|---|---|---|---|---|---|
| TN | -0.30 | **-0.40** | **-0.70** | **0.36** | **-0.54** | **0.93** |
| TN/PM1 | **-0.63** | **-0.50** | -0.02 | **0.37** | **-0.35** | 0.36 |
| $NO_3^-$-N/TN | -0.39 | -0.04 | **-0.73** | -0.26 | **-0.77** | **0.98** |
| $NH_4^+$-N/TN | 0.16 | -0.30 | **0.60** | **0.52** | **0.42** | **-0.86** |
| OrgN/TN | 0.20 | **0.38** | 0.20 | -0.33 | **0.51** | -0.71 |
| $NO_3^-$ | **-0.41** | **-0.35** | **-0.80** | -0.03 | **-0.78** | **0.96** |
| $NH_4^+$ | -0.22 | **-0.42** | **-0.61** | **0.40** | **-0.44** | 0.75 |
| OrgN | -0.26 | -0.27 | **-0.56** | 0.30 | **-0.25** | 0.71 |
| $SO_4^{2-}$ | -0.07 | **-0.38** | -0.30 | **0.51** | 0.03 | -0.57 |
| $Cl^-$ | -0.37 | -0.18 | **-0.74** | **-0.37** | **-0.74** | **0.99** |
| $O_3$ (gas) | **0.45** | 0.14 | 0.15 | -0.02 | **0.40** | -0.71 |
| $NO_2$ (gas) | **-0.53** | **-0.34** | **-0.72** | 0.20 | **-0.64** | **0.86** |
| $NO_2/NO$ (gas) | **-0.51** | -0.26 | **-0.82** | 0.14 | **-0.76** | **0.82** |
| Temp. | **0.58** | 0.30 | **0.52** | -0.21 | **0.77** | -0.43 |

*Event data are excluded from winter and year datasets.























Table 3: Spearman correlation coefficients (r) of $\delta^{13}C$ with various tracers. Only bold values are
statistically significant (p-values < 0.05).

| $\delta^{13}C$ vs. | Autumn | Winter* | Spring | Summer | Year* | Event |
|---|---|---|---|---|---|---|
| OC | **0.64** | **0.63** | **0.91** | **0.39** | **0.75** | 0.75 |
| EC | **0.61** | **0.74** | **0.88** | 0.28 | **0.84** | 0.46 |
| EC/TC | 0.06 | 0.06 | 0.13 | **-0.51** | **0.32** | -0.32 |
| TC/PM1 | -0.16 | -0.05 | **-0.40** | 0.22 | -0.09 | 0.32 |
| $NO_3^-$ | **0.74** | **0.52** | **0.71** | 0.12 | **0.76** | 0.39 |
| $NH_4^+$ | **0.84** | **0.59** | **0.80** | **0.42** | **0.66** | 0.75 |
| Oxalate | 0.34 | **0.62** | **0.71** | **0.65** | **0.25** | **0.93** |
| $SO_4^{2-}$ | **0.80** | **0.64** | **0.73** | **0.41** | **0.34** | 0.54 |
| $K^+$ | **0.84** | **0.63** | **0.70** | **0.47** | **0.76** | **0.93** |
| $Cl^-$ | **0.44** | **0.62** | **0.68** | **0.44** | **0.76** | 0.25 |
| CO (gas) | 0.21 | **0.53** | **0.60** | 0.32 | **0.37** | 0.68 |
| $O_3$ (gas) | **-0.41** | -0.26 | 0.14 | **0.66** | **-0.33** | 0.11 |
| $NO_2$ (gas) | **0.67** | 0.38 | **0.70** | 0.18 | **0.69** | 0.32 |
| $NO_2/NO$ (gas) | **0.72** | **0.65** | **0.67** | **0.68** | **0.78** | **0.96** |
| Temp. | -0.33 | **-0.35** | -0.20 | **0.39** | **-0.57** | **-0.79** |

*Event data are excluded from winter and year datasets.


















**Figures:**

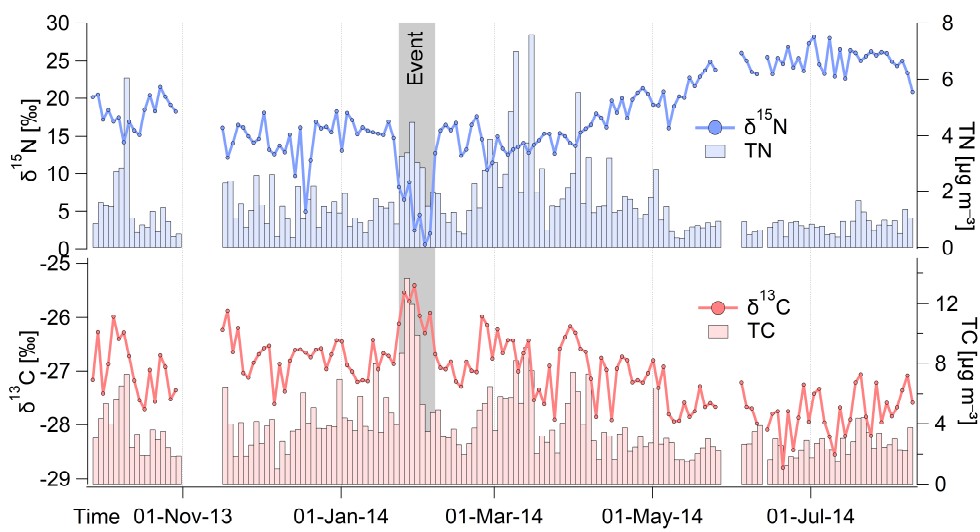


Fig. 1: Time series of $\delta^{15}$N together with TN (top) and $\delta^{13}$C together with TC (bottom) in PM1 aerosols
at the Košetice station. The gray color highlights an *Event* with divergent values, especially for $\delta^{15}$N.

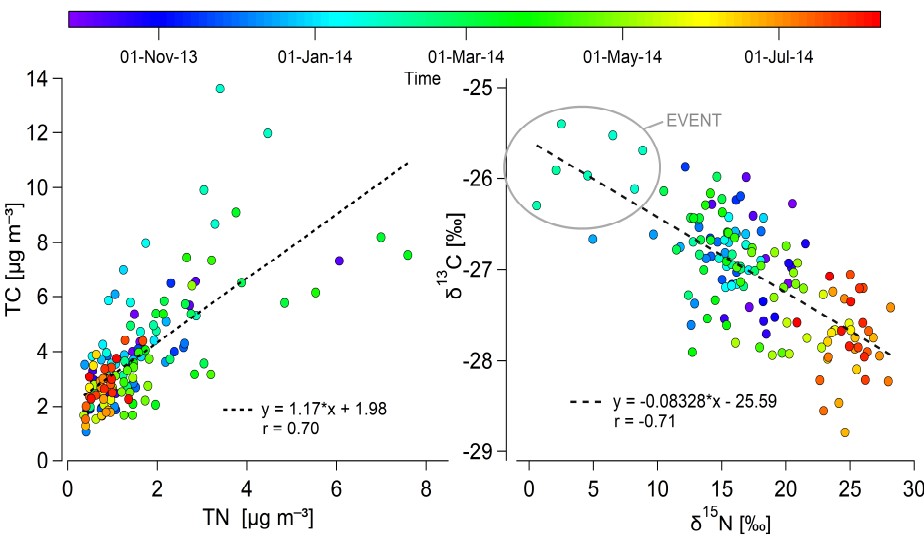


Fig. 2: Relationships between TC and TN (left) and their stable carbon and nitrogen isotopes (right).
The color scale reflects the time of sample collection. The gray circle highlights the winter *Event*
measurements.





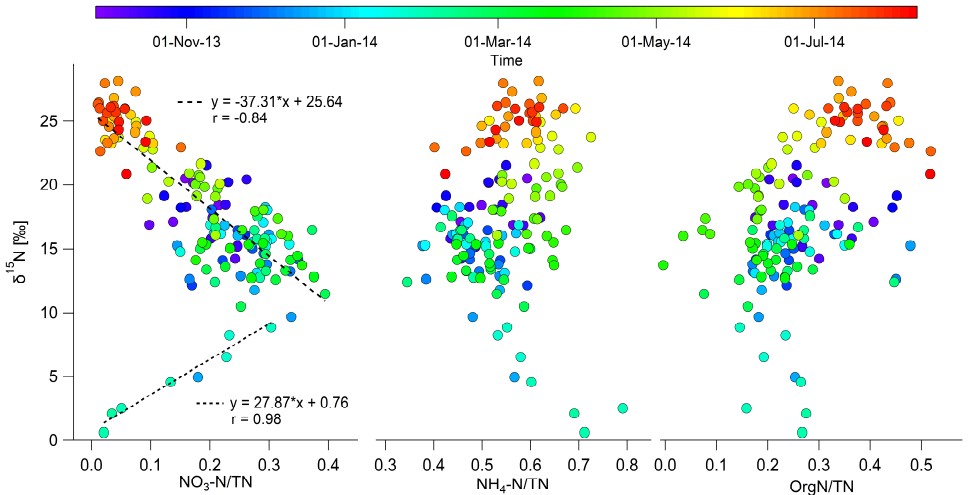

Fig. 3: Changes in $\delta^{15}$N depending on fraction of individual nitrogen components (NO$_3$-N, NH$_4$-N, and
OrgN) in TN. The color scale reflects the time of sample collection.

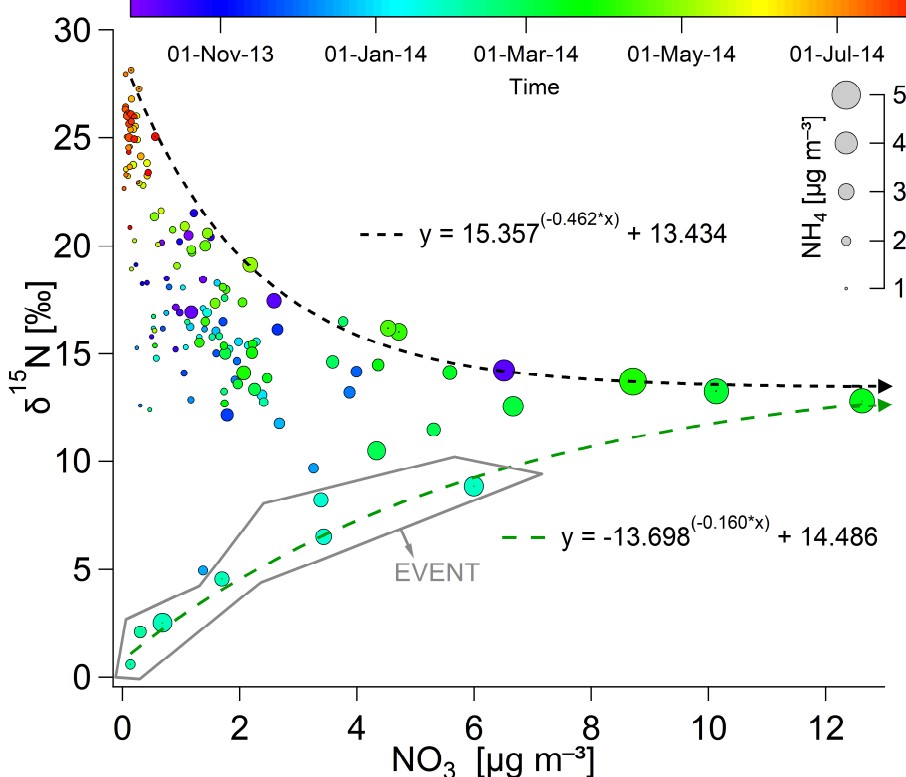

Fig. 4: Relationships of $\delta^{15}$N in TN vs. NO$_3^-$ concentrations. The larger circles indicate higher NH$_4^+$
concentrations. The color scale reflects the time of sample collection.




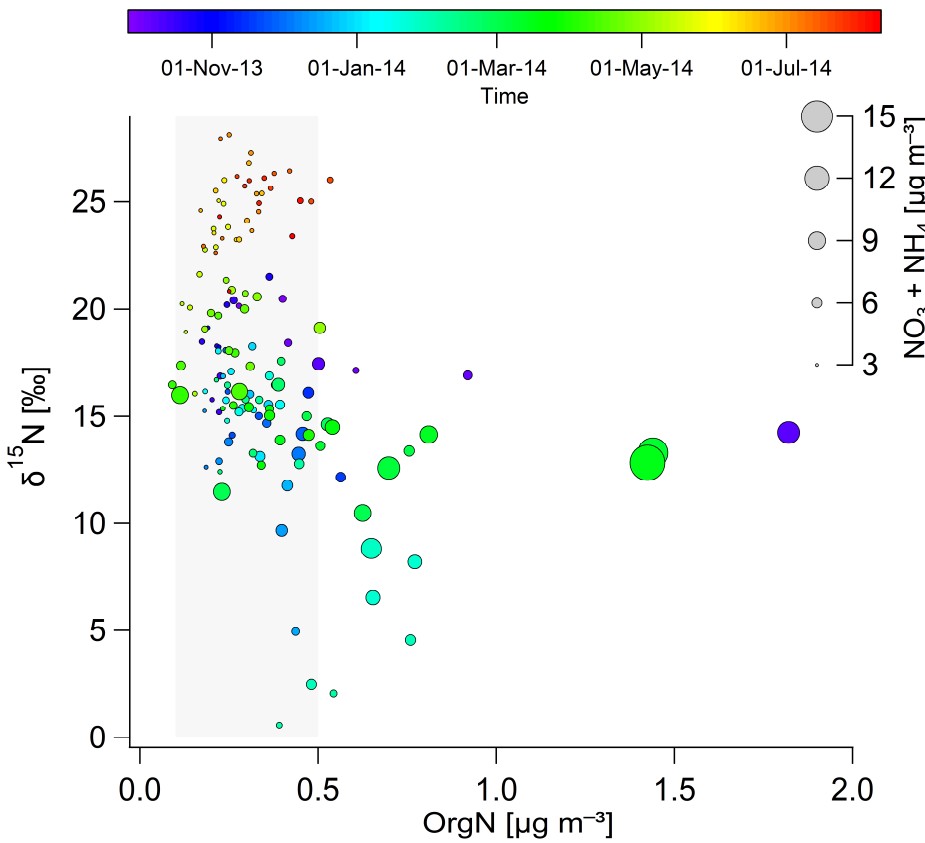


Fig. 5: Relationships of $\delta^{15}$N in TN vs. OrgN concentrations. The larger circles indicate higher sums of
NO$_3^-$+ NH$_4^+$ concentrations. The color scale reflects the time of sample collection, and the highlighted
portion is a concentration range between 0.1-0.5 µg m$^{-3}$.



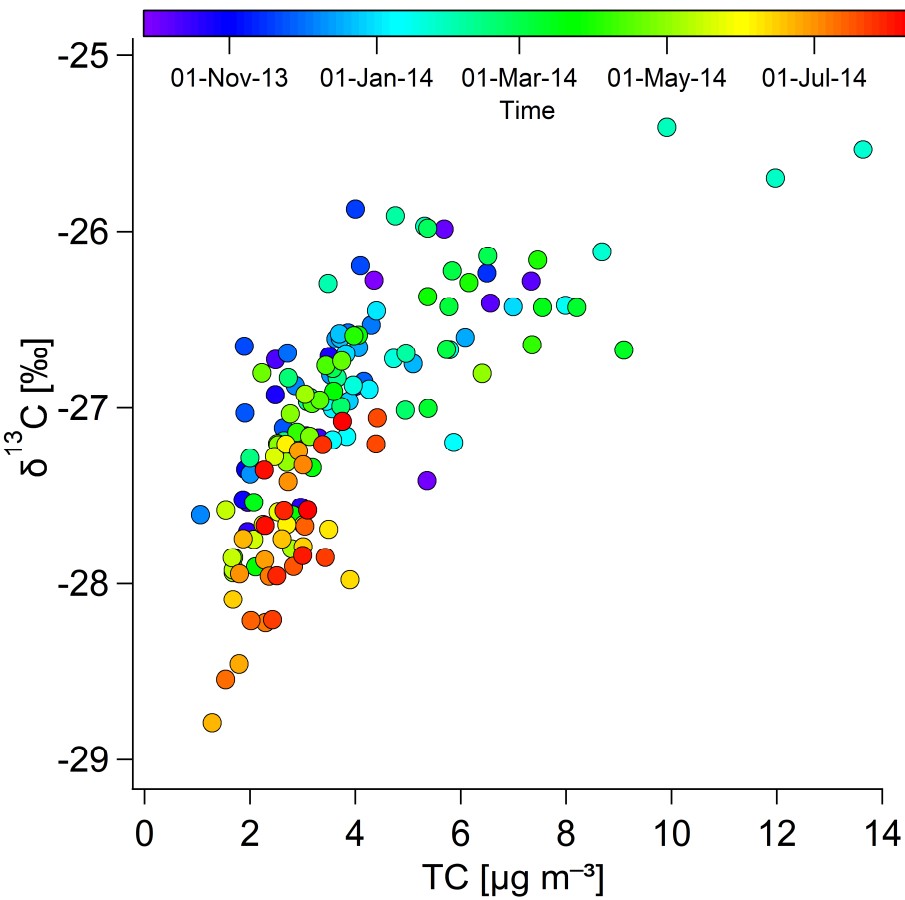

Fig. 6: Relationship between TC and δ¹³C. The color scale reflects the time of sample collection.





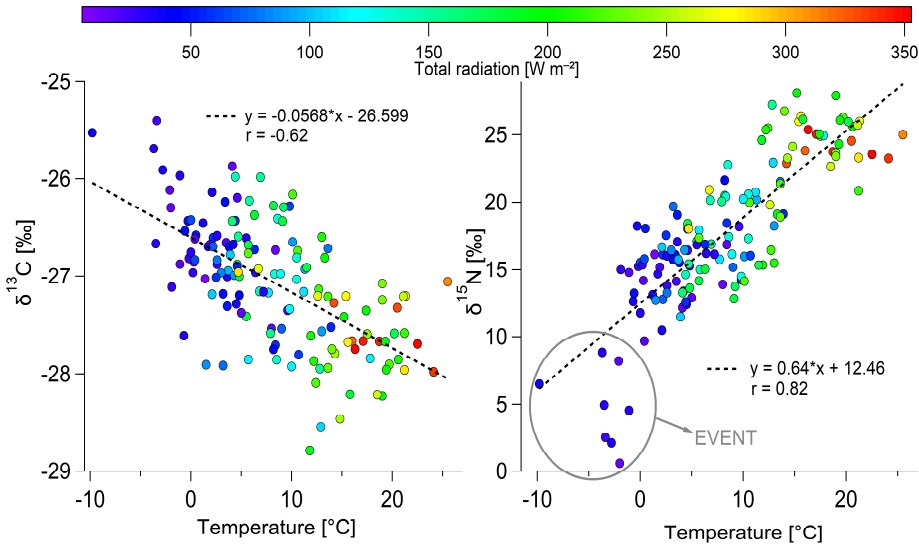


Fig. 7: Relationships between temperature and $\delta^{13}$C in TC (left) and $\delta^{15}$N in TN (right). The color scale
reflects the total radiation.



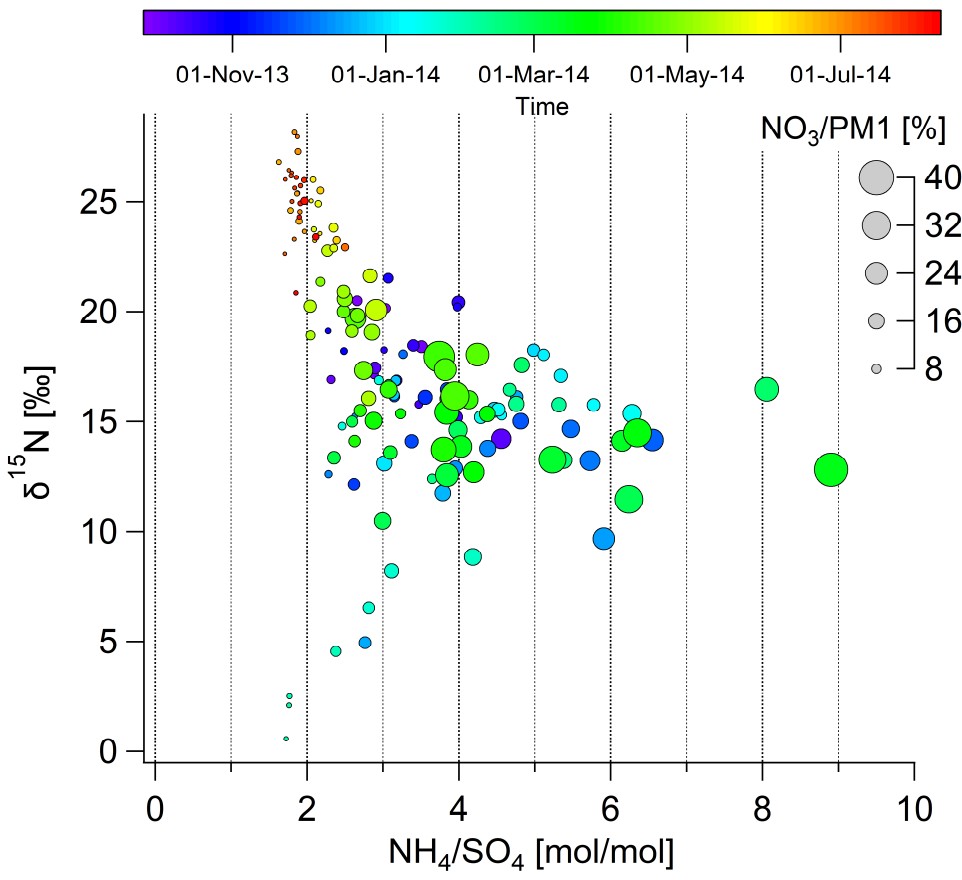

Fig. 8: Relationships between $\delta^{15}N$ in TN and molar ratios of $NH_4^+/SO_4^{2-}$ in particles. The larger circle
indicates a higher nitrate content in PM1. The color scale reflects the time of sample collection.




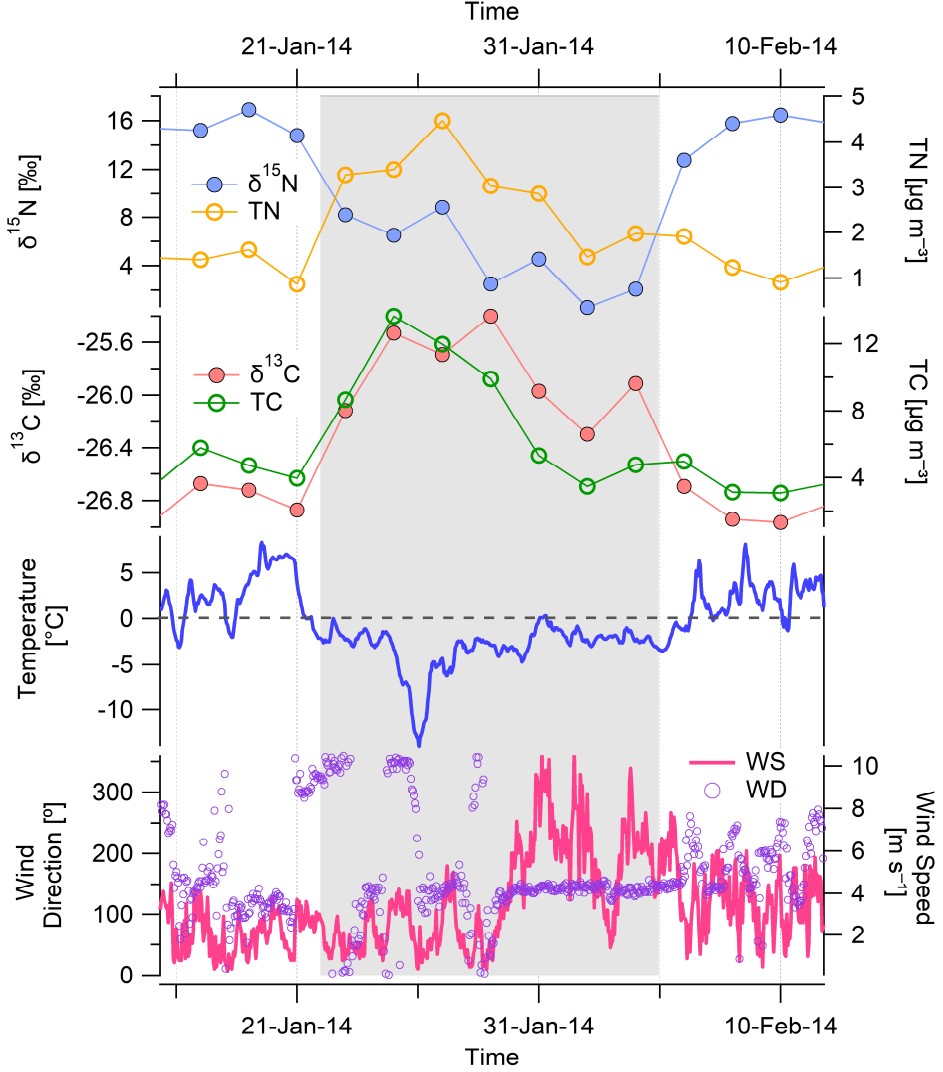

Fig. 9: Time series of $\delta^{15}N$, TN, $\delta^{13}C$, TC and meteorological variables (temperature, wind speed and
direction, 1 h time resolution) during the *Event*, which is highlighted by a gray color.



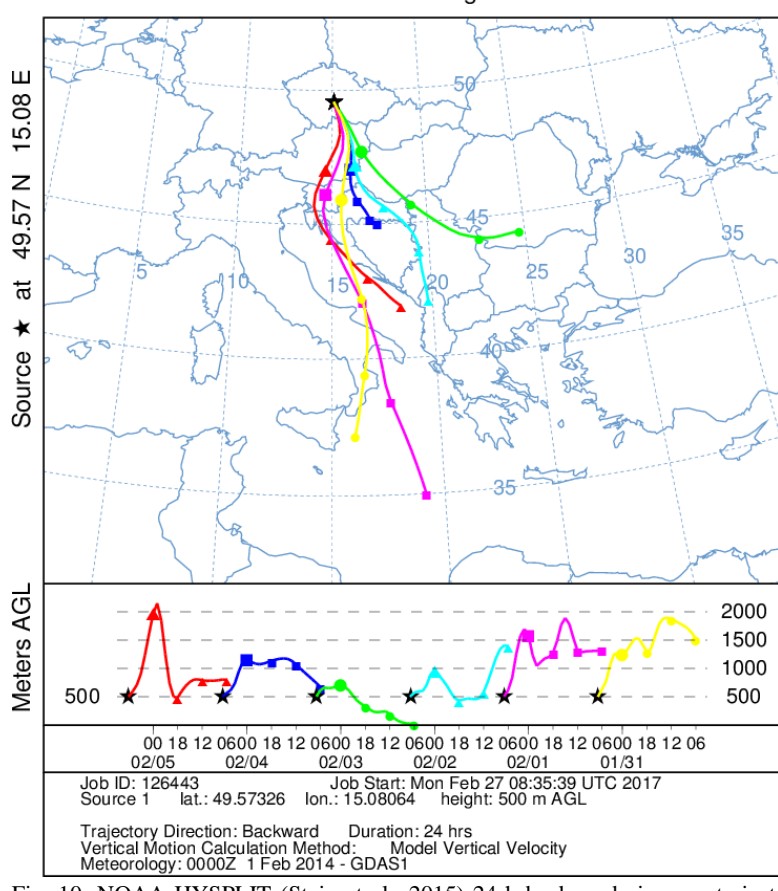

Fig. 10: NOAA HYSPLIT (Stein et al., 2015) 24 h backward air mass trajectories at 500 m above
ground level for the observation site from 30 Jan until 5 Feb 2014 (right).