# Peer review of "Seasonal study of stable carbon and nitrogen isotopic composition"

_Atmospheric Chemistry and Physics, 2018_

## Referee Comment (RC1) · Anonymous Referee #2 · 10 Oct 2018

This paper presents seasonal variations of d15N and d13C in ambient aerosol collected in Košetice (Central Europe) between 27 September 2013 and 9 August 2014. The authors show an impressive series of measurements aiming to investigate sources and processing of the fine fraction of aerosol at a rural background site. This study using two-isotope analysis is very suitable for this goal.

The use of multiple isotope ratios for the study of atmospheric pollution and the chemistry of organic compounds in the atmosphere is a newly emerging tool. The manuscript contributes to scientific progress within the scope of the journal; therefore, it is suitable to be published for discussions in ACP. Both description and discussion

of measurements are well founded. Unfortunately, the presentation is not on the same level, therefore it needs to be substantially improved before publishing.

General comments:

1) The authors discuss the benefits of using isotopes in the atmospheric research. These can give some hints to information, which is not available from concentration measurements, such as the impact of sources vs. processing on measured delta values. I miss though a discussion on the current limitations of using isotope ratio measurements for the above mentioned purpose. This omission might be the reason why the interpretation sounds sometimes so futile.

Example: Lines262-263 'In the case of our data, mixing of all of these factors probably had an influence on the nitrate isotopic composition during different parts of the year.' Reformulate!

2) The introduction should make the reader aware of the importance of using multiple isotopes (literature sources are required), e.g. for constraining potential sources. The sentence on the Lines 85-86 is too late and too less. A proper foreword would bring more structure in the discussion from Lines59-83. Here the authors must clearly differentiate between single and multiple isotope analyses.

3) Separate Spearman from Pearson correlation coefficients. For that purpose, label them for each use (e.g. in Line203).

4) Name the described variables throughout the manuscript!

Some examples: Line122: Replace 'Determination of TC, TN and their stable isotopes' by 'Determination of TC, TN concentrations and their stable isotope ratios'

Line123: Replace 'For the TC and TN analyses' by ' For the TC and TN concentration and isotopic ratio measurements'

5) Vague statements should be replaced by precise explanations throughout the paper.

[Figure]

An example: Line382: specify the 'secondary processes'

6) Generally: swap the negative numbers in ranges. The lower numbers stay first.

Examples: Line520 -40 to -28permil and Line522 -38 to -22permil

Specific comments:

Lines54-57: Reformulate! The OC/EC ratios are very different in aerosol, depending on its sources. Moreover, make more sentences of this single one. Differentiate between equilibrium and kinetic isotopic effect. Guide the reader through that by giving some information on corresponding fractionation (non-equilibrium partitioning causes much lower fractionation than chemical reactions. Contrarily, equilibrium fractionation might be significant).

Line87: No need to introduce TC and TN. It happened already in Lines12-13

Line127: I don't understand. Is the oven temperature 1000°C? How can the marble burn, if that needs 1400°C?

Line131: What does 'parts' means? Give the approximate fraction in %.

Lines135-139: Mention that the final delta values are expressed relatively to the international standards and not to the 'working' standard.

Line146: The loads on the quartz filter are meant here of course.

Lines198-200: Move these sentences to the first paragraph, they don't belong to Fig.1.

Lines218-219: Reformulate: 'but they are still in line with the linear fitting of all annual data'. This is not appropriate.

Lines290-291: Reformulate! Either state that the samples containing the highest NO3- concentration show a d15N of..., or fit a histogram plot showing a peak of measurements with NO3- concentrations higher than... at a delta value of 14+/-1 permil.

Lines300-307: The paragraph should be moved upward to Fig. 3.

Lines338-349: Completely rearrange! Suggestion: start with a statement 'The measured TC d13C ranged between.... These values are ... (in which part?) situated in the reported ranges... (here give an overall range. for that take the information from e.g. the review by gensch et al. 2014). This broad range can be explained by... (plants, marine, combustion sources... whatever). (At this point bring the similarity to other european reported values).'

Line349: Replace ' The d13C values are significantly smaller than those of d15N due to' by 'The range of TC d13C values is significantly smaller than that of TN d15N due to'

Lines358-359: This comparison is confusing: what do you mean? Similar to what? Do you refer the first or the second sentence?

Lines365-370: Change the order of these two sentences. Describe first the observations and then give the explanation.

Line 375: Replace 'these isotopes' with 'isotope distributions'.

Lines379-380: Not the changes in aerosol chemistry are different, but the chemistry itself.

Lines386-391: Change the order of the first two sentences. The third one describes the first not the second one.

Lines415-422: Lack of clarity! Reformulate, by bringing some structure in it: starting at high NH4/SO4 down to 2 and lower than 2! For each range: particle components, processes (e.g. NH3 deficit in gas phase at ratios <2), seasonal dependence.

Lines429-434: Too abrupt! Start with the observation of similar gaseous NH3 in summer and winter. Describe what a thermodynamic equilibrium would mean for the particles and how would this be reflected in the delta values. Measurements show a different situation -> more organics in summer...

Lines482-484: Very confuse sentence. Reformulate!

Lines570-574: The winter observation should stay before the summer ones. In that way, the flow is more coherent (e.g. no need to explain lower values of TN d15N when there are high fraction of nitrates.).

Editorial revisions:

The used English is not optimal. I do not give any editorial advises! My only suggestion is that this manuscript MUST be carefully revised by a native speaker. The work is too good to risk to make the reader hostile due to the language.

The manuscript is 'peppered' with:

1) Wrong prepositions

- Lines43-44 'Key processes in the atmosphere, which are involved WITH climate changes, air quality, rain events (Fuzzi et al., 2015) or visibility (Hyslop, 2009), are strongly influenced by aerosols. ' - Lines391-392 ' Although Savard et al. (2017) reported a similar negative d15N in NH4+ dependence AT temperatures in Alberta (Canada),...' Also the word order is wrong.

2) Unhandy expressions

- Lines325-328:' During the domestic heating season with the highest concentrations of NO3and NH4+, we can observe a significant increase in OrgN with $\delta$15N again at approximately 14‰ which implies that the isotopic composition of OrgN is determined by the same process during maximal NO3-concentrations, that is, emissions from domestic heating.'

3) Long, confusing sentences

Lines361-365 or Lines391-396. In these cases it helps to divide into more clear sentences.

---

## Referee Comment (RC2) · Anonymous Referee #1 · 12 Nov 2018

The paper of Vodicka et al. aimed at elucidating particulate matter and their gaseous precursor sources by interpreting results of isotope analysis. The study is based on a year-round data set and, therefore, trying to discern seasonal differences and processes taking place in different seasons. The analytical approach to the measurement results is highly commendable where authors try to make sense of various relationships between the variables. By large the interpretations are justified although several of them are highly speculative and aiming to fit the data or simply shallow. Graphics of the paper is very well prepared and clear utilising three or four dimensions in 2D graph. The paper is certainly recommended for publication in Atmospheric Chemistry and Physics, but additional work is required by removing ambiguities, speculations and

making the interpretations more coherent.

Major comments

The major comment is the lack of coherent interpretation arising from unified approach to isotope effect and fractionation processes. Physico-chemical properties are the result of quantum mechanical effects with heavier isotopes (like C13 and N15) possessing lower vibrational energy levels and making stronger bonds and vice versa for lighter isotopes. Also, lighter isotope species have lower vapour pressures resulting in faster phase transitions. Condensation of isotope-light species would make the product of lower isotope ratio while evaporation would make it higher. Formation/evaporation of ammonium nitrate is, indeed, a reversible reaction, but not necessarily in equilibrium from isotope point of view because of competition effect with sulfate for ammonium. Particulate matter products, like ammonium sulfate or nitrate only consume a small fraction of their respective gaseous species pool (1-10%) with a possible exception of nitric acid (which is a product itself). Hence, isotope-heavy product does not necessarily involve heavy precursors, but instead isotope-heavy fraction of gaseous precursors due to the above considerations.

Particles of different size ranges do not induce isotope effects, but are rather a result of chemical transformations or being produced by different emission sources, e.g. traffic produces mainly nanoparticles, while solid fuel combustion produce wide range of particle depending on the size of the source (industrial scale versus domestic).

It would very useful to consider isotope mixing approach in interpreting the results which would eliminate unnecessary associations, e.g. dC13 and EC (which is a relatively small fraction of TC). The authors, indeed, consider isotope mixing in few cases which is encouraged to do more frequently.

Lastly, not all of the observations or measurement results should be strictly interpreted as some may be spurious or based on small subset of data and highly uncertain. Insignificant correlations should not even be noted with numbers, they are meaningless.

Minor comments

Line 12. A study of stable carbon. . .

Line 15. 146 daily samples suggest 40% data coverage which is rather low for continuous sampling.

Line 17. Autumn and spring were transition periods. . .(use past tense as the study refers to the past).

Line 18. changing sources instead of different.

Line 21. "Controls" can be used when processes are exactly known. "A comparative analysis with .... has revealed major associations which enlightened about certain processes affecting isotopic composition".

Line 32. ". . .of nitrogen species.", instead of "on nitrogen level".

Line 36. The winter event has occurred in prevailing southeast air masses.

Line 43. "Aerosols have a strong impact on key processes in the atmosphere associated with climate change, air quality, rain patterns and visibility".

Line 47. Unique insights instead of information.

Line 49. Is atmospheric history any different from physical and chemical processes in the atmosphere? What is meant specifically by history?

Line 51. altered instead of influenced.

Line 52. Delete "in case of nitrogen" and "in the case of carbon".

Line 60. Americas or North&South America.

Line 63. focusing on seasonality.

Line 85. ...and to reveal undergoing chemical processes.

Line 86. to broaden the approach over the European....

Line 89. First study in the location or region? Surely authors must be certain about their location/station.

Line 95. ...observatory is a key station of the Czech..., focusing on air quality and environmental monitoring.

Line 99. with very low traffic density.

Line 110-112. Were 146 samples obtained as a result of continuous or strategic sampling? Unclear sampling strategy or low data coverage. Please explain. Perhaps "Some temporal gaps were caused by sampler maintenance or power outages resulting in 146 samples during a year-long study".

Line 113. sampled at a flow rate of 2.3 m3/h.

Line 119. The PM1 mass was measured gravimetrically with a microbalance (....) in a controlled environment (....).

Line 135. Thermo Electron Corp.

Line 136. ...for calculating TC, TN and their isotope values.

Line 152. measured, not detected.

Line 175. ...instrument response without filter exposure.

Line 184. ...for the annual dataset (139) and separately for each season and winter event.

Line 186. ...for the correlation analysis as their distinctly high concentrations and isotopic values might have affected the results.

Line 193. Statistically, 146 sample may be sufficient, indeed, but specific season(s) may not be typical, unless known to be such from previous studies.

Line 203. ..., but the relationship split during high concentration events due to divergent sources.

Line 208. ...and characteristic of significant shift in chemical composition.

Line 211. Use past tense as the study has been conducted in the past.

Line 209-211. Little difference does provide information on sources and quite contrary suggest that they were similar throughout the year in terms of BC production. The split contribution in each season may be different as suggested by isotope ratios, but overall the mixture of the sources seems to produce a steady trend.

Line 215. Do not report what was insignificant as it may mislead readers. "This result highlights significant shift in carbonaceous matter sources and corresponding isotope values in spring while during other seasons the sources were rather stable".

Line 217. Lack of correlation during particular season is due to stability of sources while the variability between samples is similar in all seasons. Authors may look at the fractional variability of isotope ratios in each season as it seems that relative variability of dC13 is a lot larger than dN15.

Line 224. ...which increased by an average of 13permile.

Line 232. What is the purpose of the formula if not solved for fractions (which is impossible given one equation and at least two unknowns).

Line 236. What does it mean similarly balanced if NO3 was higher in winter?

Line 258. (a) changes in NOx emissions

Line 262. "Considering our study, it was most likely that all of the factors contributed to a certain extent to isotopic composition throughout the year".

Line 266. In summary,.... If enrichment of N15 occurs during lowest NO3 contribution it can be inferred that NO3 is depleted in N15. Is this inference consistent through the

year?

Line 275. Size fraction has no impact if the most of nitrogen containing particles reside in submicron range. Life-time has no impact either if coarse particles do not contain appreciable amount of nitrogen. If they do, what compounds that would be and how did they end up in coarse particles. If those compounds appeared in coarse particles by condensation then nitrogen was mainly concentrated on the surface and consequently coarse particles would be as reactive as fine ones.

Line 277. Aitken mode contributes negligibly to PM1 mass making this argument very weak. Unless authors can quantitatively prove it otherwise.

Line 303. Not all of gaseous precursor mass is ending up in NH4NO3, but preferentially heavier part. Authors must consider kinetic fractionation, otherwise conclusions are biased or unfounded.

Line 317. If ratio goes up during evaporation, NH4NO3 must have had lower ratio which makes sense for highly volatile compound. One could hypothesize that ammonium N15 ratio is the same in ammonium sulfate and ammonium nitrate, but as compounds are of different volatility that is unlikely, because volatile particulate compounds originate from lighter (more depleted) precursors than less volatile compounds which originate from heavier precursors.

Line 332. "Thus it can be considered. . .

Line 342. more depleted, not "smaller".

Line 349. What does it mean "smaller"? Lower, more negative? That is not because of overlap, but source specific ratios which in case of organic carbon are largely negative.

Line 366. Wrong conclusion as mentioned in major comments. Fine particle sources are different from coarse particle sources. The ratio can only become more depleted in the atmosphere due to condensation of depleted precursors and even then condensation prefers heavier molecules, not lighter.

Line 389. "The aforementioned studies concluded that the isotope equilibrium exchange. . .

Line 397. resulting relationship, not final.

Line 401. Very narrow temperature range can produce unreliable relationships. The temperature range in this study is far more impressive.

Line 413. What is the actual process of aging? Isn't it just the production of ammonium sulfate? Sure, production is two step: first bisulfate, than sulfate. It is obvious that decreasing molar ratio corresponds to lower nitrate, because ammonium nitrate can only be produced if at least bisulfate has been produced. When nitrate is not competing for ammonium due to higher temperature, sulfate can become fully neutralized.

Line 436. Oxidation by ozone indeed makes organic matter enriched in heavier carbon, because ozone attacks unsaturated bonds and those involving lighter carbon are preferentially broken releasing "light" $CH_3$ fragments making the bulk matter heavier.

Line 444. ". . .depleted products in summer". Is this contradictory to the above paragraph?

Line 454. EC is a minor fraction of TC, so correlations are a bit pointless as EC isotope content contributes little to the TC isotope content.

Line 476. Sulfate cannot be emitted, reword.

Line 507. Dry deposition of fine particulate matter is negligible making this assertion pure speculation. Dry deposition of ammonia can only occur on surfaces - were these frosted surfaces? Confusing interpretation, please reword.

Table 2&3. Omit insignificant correlation as they are meaningless.
* * *

---

## Author Comment (AC1) · 13 Jan 2019

Response to anonymous Referee RC1

Firstly, we would like to thank the referee for his positive criticism and valuable review which has enabled us to improve our paper. Based on the reviewer comments, we thought about different views of the issue and we also re-wrote some parts of the manuscript. Answers for reviewer's comments are following below.

The paper of Vodička et al. aimed at elucidating particulate matter and their gaseous precursor sources by interpreting results of isotope analysis. The study is based on

[Figure]

a year-round data set and, therefore, trying to discern seasonal differences and processes taking place in different seasons. The analytical approach to the measurement results is highly commendable where authors try to make sense of various relationships between the variables. By large the interpretations are justified although several of them are highly speculative and aiming to fit the data or simply shallow. Graphics of the paper is very well prepared and clear utilising three or four dimensions in 2D graph. The paper is certainly recommended for publication in Atmospheric Chemistry and Physics, but additional work is required by removing ambiguities, speculations and making the interpretations more coherent.

Major comments The major comment is the lack of coherent interpretation arising from unified approach to isotope effect and fractionation processes. Physico-chemical properties are the result of quantum mechanical effects with heavier isotopes (like C13 and N15) possessing lower vibrational energy levels and making stronger bonds and vice versa for lighter isotopes. Also, lighter isotope species have lower vapour pressures resulting in faster phase transitions. Condensation of isotope-light species would make the product of lower isotope ratio while evaporation would make it higher. Formation/evaporation of ammonium nitrate is, indeed, a reversible reaction, but not necessarily in equilibrium from isotope point of view because of competition effect with sulfate for ammonium. Particulate matter products, like ammonium sulfate or nitrate only consume a small fraction of their respective gaseous species pool (1-10%) with a possible exception of nitric acid (which is a product itself). Hence, isotope-heavy product does not necessarily involve heavy precursors, but instead isotope-heavy fraction of gaseous precursors due to the above considerations.

Response: Thank you for this comment. It provides some general information that were missing in our manuscript. Based on this, we decided to extend the Introduction chapter to the following paragraph: "Isotopes are furthermore altered mainly by kinetic and/or equilibrium fractionation processes. Kinetic isotope effects (KIE) occur mainly during unidirectional (irreversible) reactions but also diffusion or during reversible reactions that are not yet at equilibrium (Gensch et al., 2014). Owing to KIE, reaction products (both gasses and particles) are depleted in the heavy isotope relatively to the reactants, and this effect is generally observed in organic compounds (Irei et al., 2006). If the partitioning between phases is caused by non-equilibrium processes (such as e.g. absorption), the isotopic fractionation is small and lower than that caused by chemical reactions (Rahn and Eiler, 2001). Equilibrium isotope effects occur in reversible chemical reactions or phase changes if the system is in equilibrium. Under such conditions, the heavier isotope is bound into the compounds where the total energy of the system is minimized and the most stable. Equilibrium effects are typical for inorganic species and usually temperature dependent."

Particles of different size ranges do not induce isotope effects, but are rather a result of chemical transformations or being produced by different emission sources, e.g. traffic produces mainly nanoparticles, while solid fuel combustion produce wide range of particle depending on the size of the source (industrial scale versus domestic).

Response: We agree with the reviewer that just different particle size itself does not induce isotope effects on these particles, and we even do not say such statement in the paper so we are sorry if some part of text sounds so. We also agree with reviewer that in first round different emission sources have effect both on particle size and isotopic composition, which is valid especially for $\delta$13C values. But moreover, size of particles has an effect on different reactivity of these particles – e.g. compounds in small particles react more often than bigger one because their effective surface for reactions is larger. Submicron particles of accumulation mode also persist longer time in atmosphere so isotopic effects also take longer time than on coarse particles and may differ in the resulting isotope composition. The above implies that even if the same isotope effects occur on the particles, the indirect properties resulting from the particle size can lead to a different isotopic composition during particle chemical/physical transformations. And this is what we try to describe in the article. For details, see answer for specific comment related to line 275 to effect of size on nitrogen isotope contents, and answer for specific comment related to line 366 due to carbon.

It would very useful to consider isotope mixing approach in interpreting the results which would eliminate unnecessary associations, e.g. dC13 and EC (which is a relatively small fraction of TC). The authors, indeed, consider isotope mixing in few cases which is encouraged to do more frequently.

Response: Regarding to associations between $\delta$13C and EC we answered in specific comment related to line 454 (see below).

Regarding to multi-isotope approach, we extended Introduction chapter about examples of studies using multi isotope analyses – see following text: "Recently, the multiple isotope approach was applied in several studies by using $\delta$13C and $\delta$15N measurements. Specifically, the $\delta$13C and $\delta$15N composition of aerosol (along with other supporting data) was used to identify the sources and processes on marine sites in Asia (Bikkina et al., 2016; Kunwar et al., 2016; Miyazaki et al., 2011; Xiao et al., 2018). Same isotopes were used to determine the contribution of biomass burning to organic aerosols in India (Boreddy et al., 2018) and in Tanzania (Mkoma et al., 2014), or to unravel the sources of aerosol contamination at Cuban rural and urban coastal sites (Morera-Gómez et al., 2018). These studies show the potential advantages of $\delta$13C and $\delta$15N isotope ratios to characterize aerosol types and to reveal the underlying chemical processes that take place in them."

Lastly, not all of the observations or measurement results should be strictly interpreted as some may be spurious or based on small subset of data and highly uncertain. Insignificant correlations should not even be noted with numbers, they are meaningless.

Response: Thank you for this comment. Actually, we had much bigger correlation matrix and we choose only correlations which were somehow interesting and also make sense for following interpretations. These value are summarized in Tables 2 and 3. Regarding statistical significance/insignificance, this is determined based on p-value. Results are statistically significant if p-value is less than 0.05 in case of our study, and

how it is used in case of many similar studies. However, the significance level of 0.05 is just a convention, and it is not entirely appropriate to omit the results of insignificant correlations just based on p-value. In fact, applying this discriminatory value is the subject of many disputations between statisticians, and using of p-value was even banned in some scientific journals (e.g. Siegfried, 2015, on-line). In some cases, the difference between statistically significant and insignificant correlation can be very small - see for example the correlation of $\delta$13C with winter temperature (-0.35, p=0.0328 => significant) and autumn temperature (-0.33, p=0.1063 => insignificant) (Table 3). In other cases, this difference can be much greater, and comparison of such differences has meaning. We agree it is not necessary to comment insignificant correlations and in this sense we removed all insignificant correlations from text. However, we would like to keep these values in Tables 2 and 3 because it can be interesting for some readers, and it can be a good compromise of this situation.

Minor comments:

Line 12. A study of stable carbon...

Response: Reworded

Line 15. 146 daily samples suggest 40% data coverage which is rather low for continuous sampling.

Response: We agree that the original formulation could be confusing. Sampling was performed every second day with 24-h time resolution and in this sense we changed this part of sentence from "...collected on a daily basis at a rural background site..." to "...collected every two days with a 24 h sampling period at a rural background site...".

Line 17. Autumn and spring were transition periods...(use past tense as the study refers to the past).

Response: Changed

Line 18. changing sources instead of different.

Response: Changed

Line 21. "Controls" can be used when processes are exactly known. "A comparative analysis with .... has revealed major associations which enlightened about certain processes affecting isotopic composition".

Response: The sentence was reformulated in sense of comment to the following text: "A comparative analysis with water-soluble ions, organic carbon, elemental carbon, trace gases and meteorological parameters (mainly ambient temperature) has shown major associations with the isotopic compositions, which enlightened the affecting processes."

Line 32. "...of nitrogen species.", instead of "on nitrogen level".

Response: Changed

Line 36. The winter event has occurred in prevailing southeast air masses.

Response: Reformulated

Line 43. "Aerosols have a strong impact on key processes in the atmosphere associated with climate change, air quality, rain patterns and visibility".

Response: First sentence of introduction was reformulated in sense of comment. Thank you.

Line 47. Unique insights instead of information.

Response: Changed

Line 49. Is atmospheric history any different from physical and chemical processes in the atmosphere? What is meant specifically by history?

Response: We didn't mean different chemical and physical processes during history of atmosphere but different chemical origin of chemical compounds in atmosphere which is "signed" e.g. by changes of $\delta 15N$ of NO3 in ice cores (see cited paper Dean et. al, 2014). In this sense we changed end of sentence to: "atmospheric composition in history".

Line 51. altered instead of influenced.

Response: Changed

Line 52. Delete "in case of nitrogen" and "in the case of carbon".

Response: These phrases were removed during rewriting of Introduction chapter.

Line 60. Americas or North&South America.

Response: Changed to "Americas"

Line 63. focusing on seasonality.

Response: Changed

Line 85. ...and to reveal undergoing chemical processes.

Response: Changed

Line 86. to broaden the approach over the European....

Response: Changed

Line 89. First study in the location or region? Surely authors must be certain about their location/station.

Response: We meant Central European region, so we changed word "location" to "region".

Line 95. ...observatory is a key station of the Czech..., focusing on air quality and environmental monitoring.

Response: Reformulated. Thank you.

[Figure]

Line 99. with very low traffic density.

Response: Changed

Line 110-112. Were 146 samples obtained as a result of continuous or strategic sampling? Unclear sampling strategy or low data coverage. Please explain. Perhaps "Some temporal gaps were caused by sampler maintenance or power outages resulting in 146 samples during a year-long study".

Response: We are sorry that our text was little bit confusing. Sampling was not made continuously day by day but every second day (it means 24 h of sampling followed by 24 h gap). Such kind of sampling was made based on strategic decision before starting of campaign. Even so we had three bigger gaps caused by sampler maintenance or power outages. We reformulated sentences on lines 110-112 to make a text clearer. Instead of "Aerosol samples (n = 146) were collected for 24 h every two days from September..." we used "Aerosol samples were collected two days for 24 h from September...". The sentence "Some gaps in sampling were caused by outages and maintenance to the sampler." was changed to "Some temporal gaps were caused by sampler maintenance or power outages resulting in 146 samples during the almost year-long study.".

Line 113. sampled at a flow rate of 2.3 m3/h.

Response: Changed to: "...operated at the flow rate at a 2.3 m3/h."

Line 119. The PM1 mass was measured gravimetrically with a microbalance (....) in a controlled environment (....).

Response: The paragraph was rephrased and shortened in a sense of comment. Thank you.

Line 135. Thermo Electron Corp.

Response: Corrected

Line 136. ...for calculating TC, TN and their isotope values.

Response: Reformulated

Line 152. measured, not detected.

Response: Changed

Line 175. ...instrument response without filter exposure.

Response: Reformulated

Line 184. ...for the annual dataset (139) and separately for each season and winter event.

Response: Corrected

Line 186. ...for the correlation analysis as their distinctly high concentrations and isotopic values might have affected the results.

Response: Sentence was completed in a sense of comment. Thank you.

Line 193. Statistically, 146 sample may be sufficient, indeed, but specific season(s) may not be typical, unless known to be such from previous studies.

Response: Seasonal comparison with other years provide e.g. work of Mbengue et al. (2018) (also cited in our work), which is published 4 years survey of EC/OC together with other variables (e.g. temperature) between years 2013-2016 directly at the Košetice station. It shows that a period from this work is not seasonally atypical during last years of observation.

Line 203. ..., but the relationship split during high concentration events due to divergent sources.

Response: Reformulated. Thank you!

Line 208. ...and characteristic of significant shift in chemical composition.

Response: End of sentence was changed in terms of comment.

Line 211. Use past tense as the study has been conducted in the past.

Response: Sentence changed to past tense

Line 209-211. Little difference does provide information on sources and quite contrary suggest that they were similar throughout the year in terms of BC production. The split contribution in each season may be different as suggested by isotope ratios, but overall the mixture of the sources seems to produce a steady trend.

Response: Station Košetice is a background site where the aerosol is more homogeneous than e.g. at an urban site, which can be reason for little seasonal differences of TC/TN ratio. However, previous studies from this station (Mbengue et al., 2018; Schwarz et al., 2016; Vodička et al., 2015) show that long-term concentrations of EC (and thus BC) are different in winter and summer, and sources are not similar throughout the year in terms of EC production. Different summer EC sources are consistent with a slightly higher summer TC/TN ratio than other seasons. In this sense, we changed the text to following: "As seen in Table 1, the seasonal averages of TC/TN ratios fluctuate, but their medians have similar values for autumn, winter and spring. The summer TC/TN value is higher (3.45) and characteristic of a significant shift in chemical composition, which is in line with previous studies at the site (Schwarz et al., 2016). However, seasonal differences in the TC/TN ratios were not as large as those in other works (e.g., Agnihotri et al., 2011), and thus, this ratio itself did not provide much information about aerosol sources."

Line 215. Do not report what was insignificant as it may mislead readers. "This result highlights significant shift in carbonaceous matter sources and corresponding isotope values in spring while during other seasons the sources were rather stable".

Response: Thank you for this comment. Instead of original sentence we used above suggested one by the reviewer. We also deleted values of insignificant correlation coefficients in text. Original sentence "However, there is a statistically significant correlation for spring only (-0.54), while in other seasons, correlations are statistically insignificant (autumn: -0.29, winter: -0.11 and summer: 0.07). This result shows that significant and related changes in the isotopic composition of nitrogen together with carbon occur especially in spring, while there are stable sources of particles during winter and summer." was changed to "However, there is a statistically significant correlation for spring only (-0.54), while in other seasons, correlations are statistically insignificant. This result highlights a significant shift in the sources of carbonaceous aerosols and their isotope values in spring while the sources were rather stable during other seasons."

Line 217. Lack of correlation during particular season is due to stability of sources while the variability between samples is similar in all seasons. Authors may look at the fractional variability of isotope ratios in each season as it seems that relative variability of dC13 is a lot larger than dN15.

Response: We agree with the review that lack of correlations during seasons is due to stability of sources in this period, and in this sense we also changed the sentence on line 215 (see previous comment). In case of reviewer suggestion related to fractional variability of isotope ratios, we are not sure by its meaning because variability of $\delta$15N is larger than variability of $\delta$13C.

Line 224. ...which increased by an average of 13permile.

Response: We changed original word "deviated" to "showed" instead of reviewer proposed "increase" because change is from 15 to 13‰ Whole sentence is following: "The $\delta$15N values are stable in winter at approximately 15‰ with an exception of the winter Event, which showed by an average of 13‰."

Line 232. What is the purpose of the formula if not solved for fractions (which is impossible given one equation and at least two unknowns).

Response: The reason for showing this formula is to give readers an idea before discussion about contribution of different nitrogen compounds to $\delta15N$ of TN and also to discussion about results presented in Fig. 3. However, because it is an equation with general information character, we decided to move this equation to the Introduction chapter.

Line 236. What does it mean similarly balanced if NO3 was higher in winter?

Response: The original sentence was: "...higher in winter, similarly balanced in spring and autumn, and very low in summer...". We wrote that similarly balanced were values in spring and autumn, not in spring and winter. However, we split this text to two following sentences to make this part of document clearer.

"...the highest in winter, and somewhat lower in spring and autumn. In summer when the dissociation of NH4NO3 plays an important role the NO3- content is very low and its nitrogen is partitioned from the aerosol phase to gas phase."

Line 258. (a) changes in NOx emissions

Response: Changed

Line 262. "Considering our study, it was most likely that all of the factors contributed to a certain extent to isotopic composition throughout the year".

Response: Original sentence was changed as follows: "In our study, it is most likely that all these factors contributed, to a certain extent, to the nitrogen isotopic composition of NO3- throughout the year."

Line 266. In summary,.... If enrichment of N15 occurs during lowest NO3 contribution it can be inferred that NO3 is depleted in N15. Is this inference consistent through the year?

Response: In our study, we observed the highest enrichment of $\delta15N$ in TN during summer when NO3 concentrations are lowest. However, during winter Event when NO3 contribution was also on the lowest level, $\delta15N$ in TN was contrariwise most depleted (see Figs.3 and 4). This Event shows that that exceptions may occur and we can't generalize. So, in our study, the inference, which you are proposing in comment, is not consistent throughout whole measurement campaign.

Line 275. Size fraction has no impact if the most of nitrogen containing particles reside in submicron range. Life-time has no impact either if coarse particles do not contain appreciable amount of nitrogen. If they do, what compounds that would be and how did they end up in coarse particles. If those compounds appeared in coarse particles by condensation then nitrogen was mainly concentrated on the surface and consequently coarse particles would be as reactive as fine ones.

Response: Nitrogen from NO3 is contained in sufficient amounts both in fine and coarse fractions (e.g. Ondráček et al., 2011; Pakkanen, 1996; Schwarz et al., 2012), not only in submicron range. As summarized by e.g. Kundu et al. (2010), coarse mode contains predominantly non-volatile nitrogen in a form of NaNO3 or Ca(NO3)2, whereas fine mode consists mainly from semi-volatile nitrogen from NH4NO3 and also in form of ammonium sulfate and bisulfate. If we have non-volatile nitrates in coarse fraction and predominantly NH4NO3 in fine fraction, where dissociation of NH4NO3 play an important role in enrichment of nitrogen, so it leads to some effect on the isotope composition depending on the particle size fraction. Yeatman et al. (2001) proposed presence of two different size-shift processes: dissociation/gas scavenging and dissolution/coagulation. Dissolution/coagulation processes appear to exhibit negative isotopic enrichment of nitrogen and shift both NH4+ and NO3 to the coarse mode, whereas dissociation/gas scavenging processes appear to exhibit positive enrichment factors. All this is supported also by the works of Mkoma et al. (2014) and Freyer (1991) who observed a higher enrichment of 15N in the fine fraction of the aerosol in comparison with coarse one. Last but not least, fine aerosol has larger particle surface/volume ratio than coarse one which can suggest higher reactivity of smaller particles. Above arguments lead us to the fact that we have in this case the opposite view than reviewer.

Line 277. Aitken mode contributes negligibly to PM1 mass making this argument very weak. Unless authors can quantitatively prove it otherwise.

Response: Thank you for this notice. We made a mistake in this part. Instead of Aitken mode, we should write Accumulation mode there because this mode contributes by a main part to PM1 mass and also persist the longest time in the atmosphere. In text, we changed word "Aitken" to "accumulation".

Line 303. Not all of gaseous precursor mass is ending up in NH4NO3, but preferentially heavier part. Authors must consider kinetic fractionation, otherwise conclusions are biased or unfounded.

Response: It is clear that during incorporation of nitrogen from gas phase to aerosol phase play a role both equilibrium and kinetic fractionation. Equilibrium fractionation is related to bond stability of nitrogen isotope whereas kinetic fractionation is related to the "speed" of isotope. First time, nitrogen incorporation is probably driven by kinetic fractionation because lighter isotopes react faster, but later heavier isotopes form a more stable bonds during equilibrium fractionation. In fine fraction of aerosol, we have nitrates almost exclusively in a form of NH4NO3 which undergo to dissociation to NH4+ and NO3- in water – this state is reversible and equilibrium fractionation is preferred in such system. Nevertheless, CiÄŹÅijka et al. (2016) suggested a possible kinetic exchange reactions between NH3 and NH4+ as one of three possible processes affecting nitrogen isotopic composition, especially for fossil fuels combustions during the heating season. Also Deng et al. (2018) reported the kinetic nitrogen fractionation factors between gaseous and aqueous ammonia with statement that, when the removal of degassed ammonia is not efficient, ammonia may dissolve back to the fluid, which may significantly shift the nitrogen isotope behavior from kinetic isotope fractionation toward equilibrium isotope fractionation. Indeed, all this suggests that kinetic fractionation is likely to affect the isotopic composition of fresh particles from combustion having lower $\delta$15N than in spring or summer, and before they are affected by equilibrium fractionation. Originally, on line 303, we did not present equilibrium fractionation as a dominant process, but we only compared known values of the isotopic composition of nitrogen oxides for different sources. Since it is difficult to determine exact contribution of these fractionations to the final value of the heavier isotope in the aerosol, we do not discuss exactly, however, we added following two sentences to consider an influence of kinetic fractionation in a first steps of gas-aerosol transformation: "Because of the only slight difference between above reported $\delta$15N of nitrogen oxides and our $\delta$15N of TN during maximal NO3- events, the isotope composition is probably influenced by the process of kinetic isotopic fractionation in fossil fuel combustion samples during heating season as referred by CiÄŹÅijka et al. (2016) as one of three possible processes."

Line 317. If ratio goes up during evaporation, NH4NO3 must have had lower ratio which makes sense for highly volatile compound. One could hypothesize that ammonium N15 ratio is the same in ammonium sulfate and ammonium nitrate, but as compounds are of different volatility that is unlikely, because volatile particulate compounds originate from lighter (more depleted) precursors than less volatile compounds which originate from heavier precursors.

Response: We agree with the reviewer that ammonium 15N ratio is NOT same in ammonium sulfate and ammonium nitrate and we didn't hypothesized opposite view in paper. On line 317 we wrote '...the dissociation process of NH4NO3 can cause an increase in 15N in TN during a period of higher ambient temperatures...' which is supposed to be okay, because evaporation of more volatile ammonia from NH4NO3 comes more easily during the higher temperature and the lighter isotope is released into the gaseous phase with a higher probability.

Line 332. "Thus it can be considered...

Response: Changed to "considered".

Line 342. more depleted, not "smaller".

Response: This part of text was removed during revisions.

Line 349. What does it mean "smaller"? Lower, more negative? That is not because of

overlap, but source specific ratios which in case of organic carbon are largely negative.

Response: We are sorry for a confused formulation that may imply that we compare the negative values of $\delta$13C in comparison with $\delta$15N. However, it is not so because we know that strongly negative $\delta$13C values originate from chosen carbon standard (PDB). Originally, we wanted to say that the range of TC $\delta$13C values is significantly smaller than a range of TN $\delta$15N values. Based on this, we changed original sentence: "The $\delta$13C values are significantly smaller than those of $\delta$15N..." to "The range of TC $\delta$13C values is significantly narrower than that of TN $\delta$15N..."

Line 366. Wrong conclusion as mentioned in major comments. Fine particle sources are different from coarse particle sources. The ratio can only become more depleted in the atmosphere due to condensation of depleted precursors and even then condensation prefers heavier molecules, not lighter.

Response: The conclusion on line 366 that fine particles have lower $\delta$13C values than coarse particles was consistent with the Masalaite et al. (2015) and SkipitytÄŮ et al. (2016) studies referred in the same sentence on line 367, but is probably inappropriately formulated. We agree with reviewer that aerosol sizes itself cannot induce isotope effect and differences are caused e.g. by different aerosol sources. In this sense, we changed the sentence on line 366 from the original "...relatively low $\delta$13C values in our range (up to -28.9‰ are caused by not only sources of TC but also a fact that fine particles are more 13C depleted in comparison with coarse particles (e.g., Masalaite et al., 2015; SkipitytÄŮ et al., 2016)." to the following: "relatively low $\delta$13C values in our range (up to -28.9‰ are caused because fine particles have lower $\delta$13C values in comparison with coarse particles probably due to different sources of TC. (e.g., Masalaite et al., 2015; SkipitytÄŮ et al., 2016)."

Line 389. "The aforementioned studies concluded that the isotope equilibrium exchange...

Response: Changed.

Line 397. resulting relationship, not final.

Response: Changed

Line 401. Very narrow temperature range can produce unreliable relationships. The temperature range in this study is far more impressive.

Response: Thank you, we agree that temperature range is one of benefit of our study. (min. in winter -9.8°C to max. in summer +25.5°C results in $\Delta$ 35.3°C). It is visible that if we account data from whole year (and so we take the full temperature range) we have stronger correlation between $\delta$15N and temperature than just for individual seasons. In cases of statistically significant seasonal correlations during autumn and spring we have following temperature ranges and correlations: autumn $\Delta T$=15.8°C , r=0.58; spring $\Delta T$=17.2°C, r=0.52. Pavuluri et al. (2010), whose work we compare, observed a strong correlation (r2 = 0.58) for a temperature range of $\Delta T$=6.1°C, which is a stronger correlation for a narrower temperature range than in our case. This gives an assumption that even during the narrower temperature range in the work of Pavuluri et al. (2010), we can get a relationship which is reliable for our comparison. We did not make any revisions in MS related to this comment.

Line 413. What is the actual process of aging? Isn't it just the production of ammonium sulfate? Sure, production is two step: first bisulfate, than sulfate. It is obvious that decreasing molar ratio corresponds to lower nitrate, because ammonium nitrate can only be produced if at least bisulfate has been produced. When nitrate is not competing for ammonium due to higher temperature, sulfate can become fully neutralized.

Response: We don't guess that process of aging is just production of ammonium sulfate without its further modifications. Surely, formation of sulfate through bisulfate is a major way, however, changes are not stopped after formation of sulfate. First, when the ammonium sulfate is in a solution the ions do not bond each other but they are in a form of NH4+ and SO42-. At the same time, NH3 from gas phase is absorbed into the droplet. During evaporation of water and part of ammonia, the lighter ammonia is evaporated more and aerosol is enriched by heavier NH4+. It implies, the older the aerosol the more 15N in ammonium sulfate. Second, as shown by recent research (Weber et al., 2016) sulfate is probably not a definitive compound that is not undergo to further changes in time. There probably exist an equilibrium between sulfate and bisulfate which can also affect subsequent changes in gas/particle partitioning of ammonia. Based on this, we added following sentences to related paragraph in subsection 3.3: "Finally, summer values of NH4+/SO42- molar ratio below 2 indicate that SO42- in aerosol particles at high summer temperatures may not be completely saturated with ammonium but it can be composed from mixture of (NH4)2SO4 and NH4HSO4 (Weber et al., 2016). The equilibrium reaction between these two forms of ammonium sulfates related to temperature oscillation during a day and due to vertical mixing of the atmosphere is a probable factor which leads to increased values of $\delta$15N in early summer."

Line 436. Oxidation by ozone indeed makes organic matter enriched in heavier carbon, because ozone attacks unsaturated bonds and those involving lighter carbon are preferentially broken releasing "light" CH3 fragments making the bulk matter heavier.

Response: Thank you for this supportive comment. At the end of sentence we added new following reference related to enrichment of 13C by photochemical processing of aqueous aerosols – see Pavuluri and Kawamura (2016) in references – and we also modified the sentence to following: "As seen in Table 3, summertime positive correlations of $\delta$13C with ozone (r=0.66) and temperature (0.39) indicate oxidation processes that can indirectly lead to an enrichment of 13C in organic aerosols that are enriched with oxalic acid (Pavuluri and Kawamura, 2016)."

Line 444. "...depleted products in summer". Is this contradictory to the above paragraph?

Response: It seems in contradictory to the above paragraph, however, these are two different things. Even if summer 13C is most depleted compared to other seasons (probably due to different sources) there is a possible indirect oxidation process for their enrichment. Based on correlation analysis, this process is relevant only in summer, however, this enrichment is not strong enough to reach average $\delta$13C values during other seasons. The time series in Fig. 1 show the lowest $\delta$13C values in a mid of June and slowly increasing enrichment of 13C during rest of summer, which also support this process. We did not make any revisions in MS related to this comment.

Line 454. EC is a minor fraction of TC, so correlations are a bit pointless as EC isotope content contributes little to the TC isotope content.

Response: It is true that EC is a minor fraction of TC, however, in case of our data EC contributes by 19% on average during all seasons, which is not negligible. Interpretations of the results related to EC are supported also by other correlations, namely between $\delta$13C and NO2, NO3- and EC/TC ratio, so we believe that it is not pointless. However, it is possible that these results can be biased by lower content of EC in TC thus we modified part of last sentence at the end of related paragraph as follows: "This result can be biased by the fact that EC constitutes on average 19% of TC during all seasons. However, it is consistent with positive correlations between $\delta$13C and gaseous NO2, as well as particulate nitrate, which is also significant in autumn to spring. This result is also supported by the negative correlation between $\delta$13C and EC/TC ratio (r=-0.51), which is significant only in summer."

Line 476. Sulfate cannot be emitted, reword.

Response: Changed to "formation of sulfates"

Line 507. Dry deposition of fine particulate matter is negligible making this assertion pure speculation. Dry deposition of ammonia can only occur on surfaces - were these frosted surfaces? Confusing interpretation, please reword.

Response: Thank you for this comment. It is right that dry deposition of fine particulate matter is negligible, however, in mentioned part of text we discussed possible dry deposition of gaseous ammonia. Moreover, given sentence is a reference to observations of Savard et al. (2017). Throughout the Event, the temperature was below 0°C (see Fig.9), so frosted surfaces could be possible on the ground surface, which can support deposition but also decrease fluxes of ammonia from e.g. water surfaces or soil. Also the results indicate that during the Event gradually rose deficit of ammonia and after some time the main source of ammonia were probably agricultural emissions from farms whose emissions of NH3 are not as affected by low temperatures. Nevertheless, because we did not measure ammonia fluxes during the campaign, the conclusions on dry deposition heavier isotope of ammonia may sound little bit speculative (as the reviewer mentioned). In this sense we changed text in subsection 3.4 and in Conclusions.

[revised manuscript text omitted]

Siegfried, T.: P value ban: small step for a journal, giant leap for science, Science News, 2015. on-line: https://www.sciencenews.org/blog/context/p-value-ban-small-step-journal-giant-leap-science?mode=blog&context=117

SkipitytÄŮ, R., MašalaitÄŮ, A., Garbaras, A., MickienÄŮ, R., RagažinskienÄŮ, O., BaliukonienÄŮ, V., Bakutis, B., ŠiugždaitÄŮ, J., Petkevičius, S., Maruška, A. S. and Remeikis, V.: Stable isotope ratio method for the characterisation of the poultry house environment, Isotopes Environ. Health Stud., 53(3), 243–260, doi:10.1080/10256016.2016.1230609, 2016.

Vodička, P., Schwarz, J., Cusack, M. and Ždímal, V.: Detailed comparison of OC/EC aerosol at an urban and a rural Czech background site during summer and winter, Sci. Total Environ., 518–519(2), 424–433, doi:10.1016/j.scitotenv.2015.03.029, 2015.

Weber, R. J., Guo, H., Russell, A. G. and Nenes, A.: High aerosol acidity despite declining atmospheric sulfate concentrations over the past 15 years, Nat. Geosci., 9(4), 282–285, doi:10.1038/ngeo2665, 2016.

Xiao, H.-W., Xiao, H.-Y., Luo, L., Zhang, Z.-Y., Huang, Q.-W., Sun, Q.-B. and Zeng, Z.: Stable carbon and nitrogen isotope compositions of bulk aerosol samples over the South China Sea, Atmos. Environ., 193, 1–10, doi:https://doi.org/10.1016/j.atmosenv.2018.09.006, 2018.

Yeatman, S. G., Spokes, L. J., Dennis, P. F. and Jickells, T. D.: Can the study of nitrogen isotopic composition in size-segregated aerosol nitrate and ammonium be used to investigate atmospheric processing mechanisms?, Atmos. Environ., 35(7), 1337–1345, doi:10.1016/S1352-2310(00)00457-X, 2001.

Please also note the supplement to this comment:
https://www.atmos-chem-phys-discuss.net/acp-2018-604/acp-2018-604-AC1-supplement.pdf

**Supplement:**

**Response to anonymous Referee RC1**

Firstly, we would like to thank the referee for his positive criticism and valuable review which has enabled us to improve our paper. Based on the reviewer comments, we thought about different views of the issue and we also re-wrote some parts of the manuscript. Answers for reviewer's comments are following below.

The paper of Vodicka et al. aimed at elucidating particulate matter and their gaseous precursor sources by interpreting results of isotope analysis. The study is based on a year-round data set and, therefore, trying to discern seasonal differences and processes taking place in different seasons. The analytical approach to the measurement results is highly commendable where authors try to make sense of various relationships between the variables. By large the interpretations are justified although several of them are highly speculative and aiming to fit the data or simply shallow. Graphics of the paper is very well prepared and clear utilising three or four dimensions in 2D graph. The paper is certainly recommended for publication in Atmospheric Chemistry and Physics, but additional work is required by removing ambiguities, speculations and making the interpretations more coherent.

Major comments

The major comment is the lack of coherent interpretation arising from unified approach to isotope effect and fractionation processes. Physico-chemical properties are the result of quantum mechanical effects with heavier isotopes (like C13 and N15) possessing lower vibrational energy levels and making stronger bonds and vice versa for lighter isotopes. Also, lighter isotope species have lower vapour pressures resulting in faster phase transitions. Condensation of isotope-light species would make the product of lower isotope ratio while evaporation would make it higher. Formation/evaporation of ammonium nitrate is, indeed, a reversible reaction, but not necessarily in equilibrium from isotope point of view because of competition effect with sulfate for ammonium. Particulate matter products, like ammonium sulfate or nitrate only consume a small fraction of their respective gaseous species pool (1-10%) with a possible exception of nitric acid (which is a product itself). Hence, isotope-heavy product does not necessarily involve heavy precursors, but instead isotope-heavy fraction of gaseous precursors due to the above considerations.

Response: Thank you for this comment. It provides some general information that were missing in our manuscript. Based on this, we decided to extend the Introduction chapter to the following paragraph: *"Isotopes are furthermore altered mainly by kinetic and/or equilibrium fractionation processes. Kinetic isotope effects (KIE) occur mainly during unidirectional (irreversible) reactions but also diffusion or during reversible reactions that are not yet at equilibrium (Gensch et al., 2014). Owing to KIE, reaction products (both gasses and particles) are depleted in the heavy isotope relatively to the reactants, and this effect is generally observed in organic compounds (Irei et al., 2006). If the partitioning between phases is caused by non-equilibrium processes (such as e.g. absorption), the isotopic fractionation is small and lower than that caused by chemical reactions (Rahn and Eiler, 2001). Equilibrium isotope effects occur in reversible chemical reactions or phase changes if the system is in equilibrium. Under such conditions, the heavier isotope is bound into the compounds where the total energy of the system is minimized and the most stable. Equilibrium effects are typical for inorganic species and usually temperature dependent."*

Particles of different size ranges do not induce isotope effects, but are rather a result of chemical transformations or being produced by different emission sources, e.g. traffic produces mainly nanoparticles, while solid fuel combustion produce wide range of particle depending on the size of the source (industrial scale versus domestic).

Response: We agree with the reviewer that just different particle size itself does not induce isotope effects on these particles, and we even do not say such statement in the paper so we are sorry if some part of text sounds so. We also agree with reviewer that in first round different emission sources have effect both on particle size and isotopic composition, which is valid especially for $\delta^{13}$C values. But moreover, size of particles has an effect on different reactivity of these particles – e.g. compounds in small particles react more often than bigger one because their effective surface for reactions is larger. Submicron particles of accumulation mode also persist longer time in atmosphere so isotopic effects also take longer time than on coarse particles and may differ in the resulting isotope composition. The above implies that even if the same isotope effects occur on the particles, the indirect properties resulting from the particle size can lead to a different isotopic composition during particle chemical/physical transformations. And this is what we try to describe in the article.

For details, see answer for specific comment related to line 275 to effect of size on nitrogen isotope contents, and answer for specific comment related to line 366 due to carbon.

It would very useful to consider isotope mixing approach in interpreting the results which would eliminate unnecessary associations, e.g. dC13 and EC (which is a relatively small fraction of TC). The authors, indeed, consider isotope mixing in few cases which is encouraged to do more frequently.

Response:  Regarding to associations between $\delta^{13}$C and EC we answered in specific comment related to line 454 (see below).

Regarding to multi-isotope approach, we extended Introduction chapter about examples of studies using multi isotope analyses – see following text:

*"Recently, the multiple isotope approach was applied in several studies by using $\delta^{13}$C and $\delta^{15}$N measurements. Specifically, the $\delta^{13}$C and $\delta^{15}$N composition of aerosol (along with other supporting data) was used to identify the sources and processes on marine sites in Asia (Bikkina et al., 2016; Kunwar et al., 2016; Miyazaki et al., 2011; Xiao et al., 2018). Same isotopes were used to determine the contribution of biomass burning to organic aerosols in India (Boreddy et al., 2018) and in Tanzania (Mkoma et al., 2014), or to unravel the sources of aerosol contamination at Cuban rural and urban coastal sites (Morera-Gómez et al., 2018). These studies show the potential advantages of $\delta^{13}$C and $\delta^{15}$N isotope ratios to characterize aerosol types and to reveal the underlying chemical processes that take place in them."*

Lastly, not all of the observations or measurement results should be strictly interpreted as some may be spurious or based on small subset of data and highly uncertain. Insignificant correlations should not even be noted with numbers, they are meaningless.

A: Thank you for this comment. Actually, we had much bigger correlation matrix and we choose only correlations which were somehow interesting and also make sense for following interpretations. These value are summarized in Tables 2 and 3.

Regarding statistical significance/insignificance, this is determined based on p-value. Results are statistically significant if p-value is less than 0.05 in case of our study, and how it is used in case of many similar studies. However, the significance level of 0.05 is just a convention, and it is not entirely appropriate to omit the results of insignificant correlations just based on p-value. In fact, applying this discriminatory value is the subject of many disputations between statisticians, and using of p-value was even banned in some scientific journals (e.g. Siegfried, 2015, on-line).

In some cases, the difference between statistically significant and insignificant correlation can be very small - see for example the correlation of $\delta^{13}C$ with winter temperature (-0.35, p=0.0328 => significant) and autumn temperature (-0.33, p=0.1063 => insignificant) (Table 3). In other cases, this difference can be much greater, and comparison of such differences has meaning. We agree it is not necessary to comment insignificant correlations and in this sense we removed all insignificant correlations from text. However, we would like to keep these values in Tables 2 and 3 because it can be interesting for some readers, and it can be a good compromise of this situation.

Minor comments

Line 12. A study of stable carbon...

Response: Reworded

Line 15. 146 daily samples suggest 40% data coverage which is rather low for continuous sampling.

Response: We agree that the original formulation could be confusing. Sampling was performed every second day with 24-h time resolution and in this sense we changed this part of sentence from *"…collected on a daily basis at a rural background site…"* to *"…collected every two days with a 24 h sampling period at a rural background site… "*.

Line 17. Autumn and spring were transition periods...(use past tense as the study refers to the past).

Response: Changed

Line 18. changing sources instead of different.

Response: Changed

Line 21. "Controls" can be used when processes are exactly known. "A comparative analysis with .... has revealed major associations which enlightened about certain processes affecting isotopic composition".

Response: The sentence was reformulated in sense of comment to the following text:
*"A comparative analysis with water-soluble ions, organic carbon, elemental carbon, trace gases and meteorological parameters (mainly ambient temperature) has shown major associations with the isotopic compositions, which enlightened the affecting processes."*

Line 32. "...of nitrogen species.", instead of "on nitrogen level".

Response:  Changed

Line 36. The winter event has occurred in prevailing southeast air masses.

Response: Reformulated

Line 43. "Aerosols have a strong impact on key processes in the atmosphere associated with climate change, air quality, rain patterns and visibility".

Response: First sentence of introduction was reformulated in sense of comment. Thank you.

Line 47. Unique insights instead of information.

Response: Changed

Line 49. Is atmospheric history any different from physical and chemical processes in the atmosphere? What is meant specifically by history?

Response: We didn't mean different chemical and physical processes during history of atmosphere but different chemical origin of chemical compounds in atmosphere which is "signed" e.g. by changes of $\delta^{15}N$ of NO3 in ice cores (see cited paper Dean et. al, 2014).
In this sense we changed end of sentence to:  "*atmospheric composition in history*".

Line 51. altered instead of influenced.

Response: Changed

Line 52. Delete "in case of nitrogen" and "in the case of carbon".

Response:  These phrases were removed during rewriting of Introduction chapter.

Line 60. Americas or North&South America.

Response: Changed to "Americas"

Line 63. focusing on seasonality.

Response: Changed

Line 85. ...and to reveal undergoing chemical processes.

Response: Changed

Line 86. to broaden the approach over the European....

Response: Changed

Line 89. First study in the location or region? Surely authors must be certain about their location/station.

Response: We meant Central European region, so we changed word *"location"* to *"region"*.

Line 95. ...observatory is a key station of the Czech..., focusing on air quality and environmental monitoring.

Response: Reformulated. Thank you.

Line 99. with very low traffic density.

Response: Changed

Line 110-112. Were 146 samples obtained as a result of continuous or strategic sampling? Unclear sampling strategy or low data coverage. Please explain. Perhaps "Some temporal gaps were caused by sampler maintenance or power outages resulting in 146 samples during a year-long study".

Response: We are sorry that our text was little bit confusing. Sampling was not made continuously day by day but every second day (it means 24 h of sampling followed by 24 h gap). Such kind of sampling was made based on strategic decision before starting of campaign. Even so we had three bigger gaps caused by sampler maintenance or power outages.
We reformulated sentences on lines 110-112 to make a text clearer. Instead of "*Aerosol samples (n = 146) were collected for 24 h every two days from September…*" we used "*Aerosol samples were collected two days for 24 h from September…*".
The sentence "*Some gaps in sampling were caused by outages and maintenance to the sampler.*" was changed to "*Some temporal gaps were caused by sampler maintenance or power outages resulting in 146 samples during the almost year-long study.*".

Line 113. sampled at a flow rate of 2.3 m3/h.

Response: Changed to: *"...operated at the flow rate at a 2.3 m$^3$/h."*

Line 119. The PM1 mass was measured gravimetrically with a microbalance (....) in a controlled environment (....).

Response: The paragraph was rephrased and shortened in a sense of comment. Thank you.

Line 135. Thermo Electron Corp.

Response: Corrected

Line 136. ...for calculating TC, TN and their isotope values.

Response: Reformulated

Line 152. measured, not detected.

Response: Changed

Line 175. ...instrument response without filter exposure.

Response: Reformulated

Line 184. ...for the annual dataset (139) and separately for each season and winter event.

Response: Corrected

Line 186. ...for the correlation analysis as their distinctly high concentrations and isotopic values might have affected the results.

Response: Sentence was completed in a sense of comment. Thank you.

Line 193. Statistically, 146 sample may be sufficient, indeed, but specific season(s) may not be typical, unless known to be such from previous studies.

Response: Seasonal comparison with other years provide e.g. work of Mbengue et al. (2018) (also cited in our work), which is published 4 years survey of EC/OC together with other variables (e.g. temperature) between years 2013-2016 directly at the Košetice station. It shows that a period from this work is not seasonally atypical during last years of observation.

Line 203. ..., but the relationship split during high concentration events due to divergent sources.

Response: Reformulated. Thank you!

Line 208. ...and characteristic of significant shift in chemical composition.

Response: End of sentence was changed in terms of comment.

Line 211. Use past tense as the study has been conducted in the past.

Response: Sentence changed to past tense

Line 209-211. Little difference does provide information on sources and quite contrary suggest that they were similar throughout the year in terms of BC production. The split contribution in each season may be different as suggested by isotope ratios, but overall the mixture of the sources seems to produce a steady trend.

Response: Station Košetice is a background site where the aerosol is more homogeneous than e.g. at an urban site, which can be reason for little seasonal differences of TC/TN ratio. However, previous studies from this station (Mbengue et al., 2018; Schwarz et al., 2016; Vodička et al., 2015) show that long-term concentrations of EC (and thus BC) are different in winter and summer, and sources are not similar throughout the year in terms of EC production. Different summer EC sources are consistent with a slightly higher summer TC/TN ratio than other seasons. In this sense, we changed the text to following:

*"As seen in Table 1, the seasonal averages of TC/TN ratios fluctuate, but their medians have similar values for autumn, winter and spring. The summer TC/TN value is higher (3.45) and characteristic of a significant shift in chemical composition, which is in line  with previous studies at the site* (Schwarz et al., 2016)*. However, seasonal differences in the TC/TN ratios were not as large as those in other works (e.g., Agnihotri et al., 2011), and thus, this ratio itself did not provide much information about aerosol sources."*

Line 215. Do not report what was insignificant as it may mislead readers. "This result highlights significant shift in carbonaceous matter sources and corresponding isotope values in spring while during other seasons the sources were rather stable".

Response: Thank you for this comment. Instead of original sentence we used above suggested one by the reviewer. We also deleted values of insignificant correlation coefficients in text.
Original sentence
*"However, there is a statistically significant correlation for spring only (-0.54), while in other seasons, correlations are statistically insignificant (autumn: -0.29, winter: -0.11 and summer: 0.07). This result shows that significant and related changes in the isotopic composition of nitrogen together with carbon occur especially in spring, while there are stable sources of particles during winter and summer."*
was changed to

*"However, there is a statistically significant correlation for spring only (-0.54), while in other seasons, correlations are statistically insignificant. This result highlights a significant shift in the sources of carbonaceous aerosols and their isotope values in spring while the sources were rather stable during other seasons."*

Line 217. Lack of correlation during particular season is due to stability of sources while the variability between samples is similar in all seasons. Authors may look at the fractional variability of isotope ratios in each season as it seems that relative variability of dC13 is a lot larger than dN15.

Response: We agree with the review that lack of correlations during seasons is due to stability of sources in this period, and in this sense we also changed the sentence on line 215 (see previous comment).
In case of reviewer suggestion related to fractional variability of isotope ratios, we are not sure by its meaning because variability of $\delta^{15}N$ is larger than variability of $\delta^{13}C$.

Line 224. ...which increased by an average of 13permile.

Response: We changed original word *"deviated"* to *"showed"* instead of reviewer proposed *"increase"* because change is from 15 to 13‰. Whole sentence is following: *"The $\delta^{15}N$ values are stable in winter at approximately 15‰, with an exception of the winter Event, which showed by an average of 13‰."*

Line 232. What is the purpose of the formula if not solved for fractions (which is impossible given one equation and at least two unknowns).

Response: The reason for showing this formula is to give readers an idea before discussion about contribution of different nitrogen compounds to $\delta^{15}N$ of TN and also to discussion about results presented in Fig. 3. However, because it is an equation with general information character, we decided to move this equation to the Introduction chapter.

Line 236. What does it mean similarly balanced if NO3 was higher in winter?

Response: The original sentence was: *"…higher in winter, similarly balanced in spring and autumn, and very low in summer…"*. We wrote that similarly balanced were values in spring and autumn, not in spring and winter. However, we split this text to two following sentences to make this part of document clearer.

*"…the highest in winter, and somewhat lower in spring and autumn. In summer when the dissociation of $NH_4NO_3$ plays an important role the $NO_3^-$ content is very low and its nitrogen is partitioned from the aerosol phase to gas phase."*

Line 258. (a) changes in NOx emissions

Response: Changed

Line 262. "Considering our study, it was most likely that all of the factors contributed to a certain extent to isotopic composition throughout the year".

Response: Original sentence was changed as follows: *"In our study, it is most likely that all these factors contributed, to a certain extent, to the nitrogen isotopic composition of $NO_3^-$ throughout the year."*

Line 266. In summary,.... If enrichment of N15 occurs during lowest NO3 contribution it can be inferred that NO3 is depleted in N15. Is this inference consistent through the year?

Response: In our study, we observed the highest enrichment of $\delta^{15}N$ in TN during summer when $NO_3$ concentrations are lowest. However, during winter Event when $NO_3$ contribution was also on the lowest level, $\delta^{15}N$ in TN was contrariwise most depleted (see Figs.3 and 4). This Event shows that that exceptions may occur and we can't generalize. So, in our study, the inference, which you are proposing in comment, is not consistent throughout whole measurement campaign.

Line 275. Size fraction has no impact if the most of nitrogen containing particles reside in submicron range. Life-time has no impact either if coarse particles do not contain appreciable amount of nitrogen. If they do, what compounds that would be and how did they end up in coarse particles. If those compounds appeared in coarse particles by condensation then nitrogen was mainly concentrated on the surface and consequently coarse particles would be as reactive as fine ones.

Response: Nitrogen from $NO_3$ is contained in sufficient amounts both in fine and coarse fractions (e.g. Ondráček et al., 2011; Pakkanen, 1996; Schwarz et al., 2012), not only in submicron range. As summarized by e.g. Kundu et al. (2010), coarse mode contains predominantly non-volatile nitrogen in a form of $NaNO_3$ or $Ca(NO_3)_2$, whereas fine mode consists mainly from semi-volatile nitrogen from $NH_4NO_3$ and also in form of ammonium sulfate and bisulfate. If we have non-volatile nitrates in coarse fraction and predominantly $NH_4NO_3$ in fine fraction, where dissociation of $NH_4NO_3$ play an important role in enrichment of nitrogen, so it leads to some effect on the isotope composition depending on the particle size fraction. Yeatman et al. (2001) proposed presence of two different size-shift processes: dissociation/gas scavenging and dissolution/coagulation. Dissolution/coagulation processes appear to exhibit negative isotopic enrichment of nitrogen and shift both $NH_4^+$ and $NO_3$ to the coarse mode, whereas dissociation/gas scavenging processes appear to exhibit positive enrichment factors. All this is supported also by the works of Mkoma et al. (2014) and Freyer (1991) who observed a higher enrichment of $^{15}N$ in the fine fraction of the aerosol in comparison with coarse one. Last but not least, fine aerosol has larger particle surface/volume ratio than coarse one which can suggest higher reactivity of smaller particles.
Above arguments lead us to the fact that we have in this case the opposite view than reviewer.

Line 277. Aitken mode contributes negligibly to PM1 mass making this argument very weak. Unless authors can quantitatively prove it otherwise.

Response: Thank you for this notice. We made a mistake in this part. Instead of Aitken mode, we should write Accumulation mode there because this mode contributes by a main part to PM1 mass and also persist the longest time in the atmosphere. In text, we changed word *"Aitken"* to *"accumulation"*.

Line 303. Not all of gaseous precursor mass is ending up in NH4NO3, but preferentially heavier part. Authors must consider kinetic fractionation, otherwise conclusions are biased or unfounded.

Response: It is clear that during incorporation of nitrogen from gas phase to aerosol phase play a role both equilibrium and kinetic fractionation. Equilibrium fractionation is related to bond stability of nitrogen isotope whereas kinetic fractionation is related to the "speed" of isotope. First time, nitrogen incorporation is probably driven by kinetic fractionation because lighter isotopes react faster, but later heavier isotopes form a more stable bonds during equilibrium fractionation. In fine fraction of aerosol, we have nitrates almost exclusively in a form of $NH_4NO_3$ which undergo to dissociation to $NH_4^+$ and $NO_3^-$ in water – this state is reversible and equilibrium fractionation is preferred in such system. Nevertheless, Cieżka et al. (2016) suggested a possible kinetic exchange reactions between $NH_3$ and $NH_4^+$ as one of three possible processes affecting nitrogen isotopic composition, especially for fossil fuels combustions during the heating season. Also Deng et al. (2018) reported the kinetic nitrogen fractionation factors between gaseous and aqueous ammonia with statement that, when the removal of degassed ammonia is not efficient, ammonia may dissolve back to the fluid, which may significantly shift the nitrogen isotope behavior from kinetic isotope fractionation toward equilibrium isotope fractionation. Indeed, all this suggests that kinetic fractionation is likely to affect the isotopic composition of fresh particles from combustion having lower $\delta^{15}N$ than in spring or summer, and before they are affected by equilibrium fractionation.
Originally, on line 303, we did not present equilibrium fractionation as a dominant process, but we only compared known values of the isotopic composition of nitrogen oxides for different sources.
Since it is difficult to determine exact contribution of these fractionations to the final value of the heavier isotope in the aerosol, we do not discuss exactly, however, we added following two sentences to consider an influence of kinetic fractionation in a first steps of gas-aerosol transformation: *"Because of the only slight difference between above reported $\delta^{15}N$ of nitrogen oxides and our $\delta^{15}N$ of TN during maximal $NO_3^-$ events, the isotope composition is probably influenced by the process of kinetic isotopic fractionation in fossil fuel combustion samples during heating season as referred by Cieżka et al. (2016) as one of three possible processes."*

Line 317. If ratio goes up during evaporation, NH4NO3 must have had lower ratio which makes sense for highly volatile compound. One could hypothesize that ammonium N15 ratio is the same in ammonium sulfate and ammonium nitrate, but as compounds are of different volatility that is unlikely, because volatile particulate compounds originate from lighter (more depleted) precursors than less volatile compounds which originate from heavier precursors.

Response: We agree with the reviewer that ammonium [15]N ratio is NOT same in ammonium sulfate and ammonium nitrate and we didn't hypothesized opposite view in paper. On line 317 we wrote *'...the dissociation process of $NH_4NO_3$ can cause an increase in $^{15}N$ in TN during a period of higher ambient temperatures...'* which is supposed to be okay, because evaporation of more volatile ammonia from $NH_4NO_3$ comes more easily during the higher temperature and the lighter isotope is released into the gaseous phase with a higher probability.

Line 332. "Thus it can be considered...

Response:  Changed to *"considered"*.

Line 342. more depleted, not "smaller".

Response:  This part of text was removed during revisions.

Line 349. What does it mean "smaller"? Lower, more negative? That is not because of overlap, but source specific ratios which in case of organic carbon are largely negative.

Response: We are sorry for a confused formulation that may imply that we compare the negative values of $\delta^{13}C$ in comparison with $\delta^{15}N$. However, it is not so because we know that strongly negative $\delta^{13}C$ values originate from chosen carbon standard (PDB). Originally, we wanted to say that the range of TC $\delta^{13}C$ values is significantly smaller than a range of TN $\delta^{15}N$ values.
Based on this, we changed original sentence: *"The $\delta^{13}C$ values are significantly smaller than those of $\delta^{15}N...$"* to  *"The range of TC $\delta^{13}C$ values is significantly narrower than that of TN $\delta^{15}N...$"*

Line 366. Wrong conclusion as mentioned in major comments. Fine particle sources are different from coarse particle sources. The ratio can only become more depleted in the atmosphere due to condensation of depleted precursors and even then condensation prefers heavier molecules, not lighter.

Response: The conclusion on line 366 that fine particles have lower $\delta^{13}C$ values than coarse particles was consistent with the Masalaite et al. (2015) and Skipitytė et al. (2016) studies referred in the same sentence on line 367, but is probably inappropriately formulated. We agree with reviewer that aerosol sizes itself cannot induce isotope effect and differences are caused e.g. by different aerosol sources. In this sense, we changed the sentence on line 366 from the original  *"…relatively low $\delta^{13}C$ values in our range (up to -28.9‰) are caused by not only sources of TC but also a fact that fine particles are more $^{13}C$ depleted in comparison with coarse particles (e.g., Masalaite et al., 2015; Skipitytė et al., 2016)."*  to the following: *"relatively low $\delta^{13}C$ values in our range (up to -28.9‰) are caused because fine particles have lower $\delta^{13}C$ values in comparison with coarse particles probably due to different sources of TC. (e.g., Masalaite et al., 2015; Skipitytė et al., 2016)."*

Line 389. "The aforementioned studies concluded that the isotope equilibrium exchange...

Response: Changed.

Line 397. resulting relationship, not final.

Response: Changed

Line 401. Very narrow temperature range can produce unreliable relationships. The temperature range in this study is far more impressive.

Response: Thank you, we agree that temperature range is one of benefit of our study. (min. in winter -9.8°C to max. in summer +25.5°C results in Δ 35.3°C). It is visible that if we account data from whole year (and so we take the full temperature range) we have stronger correlation between $δ^{15}N$ and temperature than just for individual seasons.
In cases of statistically significant seasonal correlations during autumn and spring we have following temperature ranges and correlations: autumn ΔT=15.8°C , r=0.58; spring ΔT=17.2°C, r=0.52. Pavuluri et al. (2010), whose work we compare, observed a strong correlation ($r^2$ = 0.58) for a temperature range of ΔT=6.1°C, which is a stronger correlation for a narrower temperature range than in our case. This gives an assumption that even during the narrower temperature range in the work of Pavuluri et al. (2010), we can get a relationship which is reliable for our comparison.
We did not make any revisions in MS related to this comment.

Line 413. What is the actual process of aging? Isn't it just the production of ammonium sulfate? Sure, production is two step: first bisulfate, than sulfate. It is obvious that decreasing molar ratio corresponds to lower nitrate, because ammonium nitrate can only be produced if at least bisulfate has been produced. When nitrate is not competing for ammonium due to higher temperature, sulfate can become fully neutralized.

Response: We don't guess that process of aging is just production of ammonium sulfate without its further modifications. Surely, formation of sulfate through bisulfate is a major way, however, changes are not stopped after formation of sulfate. First, when the ammonium sulfate is in a solution the ions do not bond each other but they are in a form of $NH_4^+$ and $SO_4^{2-}$. At the same time, $NH_3$ from gas phase is absorbed into the droplet. During evaporation of water and part of ammonia, the lighter ammonia is evaporated more and aerosol is enriched by heavier $NH_4^+$. It implies, the older the aerosol the more $^{15}N$ in ammonium sulfate. Second, as shown by recent research (Weber et al., 2016) sulfate is probably not a definitive compound that is not undergo to further changes in time. There probably exist an equilibrium between sulfate and bisulfate which can also affect subsequent changes in gas/particle partitioning of ammonia. Based on this, we added following sentences to related paragraph in subsection 3.3:
*"Finally, summer values of $NH_4^+/SO_4^{2-}$ molar ratio below 2 indicate that $SO_4^{2-}$ in aerosol particles at high summer temperatures may not be completely saturated with ammonium but it can be composed from mixture of $(NH_4)_2SO_4$ and $NH_4HSO_4$ (Weber et al., 2016). The equilibrium reaction between these two forms of ammonium sulfates related to temperature oscillation during a day and due to vertical mixing of the atmosphere is a probable factor which leads to increased values of $δ^{15}N$ in early summer."*

Line 436. Oxidation by ozone indeed makes organic matter enriched in heavier carbon, because ozone attacks unsaturated bonds and those involving lighter carbon are preferentially broken releasing "light" CH3 fragments making the bulk matter heavier.

Response: Thank you for this supportive comment. At the end of sentence we added new following reference related to enrichment of $^{13}C$ by photochemical processing of aqueous aerosols – see

Pavuluri and Kawamura (2016) in references – and we also modified the sentence to following: *"As seen in Table 3, summertime positive correlations of $\delta^{13}C$ with ozone (r=0.66) and temperature (0.39) indicate oxidation processes that can indirectly lead to an enrichment of $^{13}C$ in organic aerosols that are enriched with oxalic acid (Pavuluri and Kawamura, 2016)."*

Line 444. "...depleted products in summer". Is this contradictory to the above paragraph?

Response: It seems in contradictory to the above paragraph, however, these are two different things. Even if summer $^{13}C$ is most depleted compared to other seasons (probably due to different sources) there is a possible indirect oxidation process for their enrichment. Based on correlation analysis, this process is relevant only in summer, however, this enrichment is not strong enough to reach average $\delta^{13}C$ values during other seasons. The time series in Fig. 1 show the lowest $\delta^{13}C$ values in a mid of June and slowly increasing enrichment of $^{13}C$ during rest of summer, which also support this process. We did not make any revisions in MS related to this comment.

Line 454. EC is a minor fraction of TC, so correlations are a bit pointless as EC isotope content contributes little to the TC isotope content.

Response: It is true that EC is a minor fraction of TC, however, in case of our data EC contributes by 19% on average during all seasons, which is not negligible. Interpretations of the results related to EC are supported also by other correlations, namely between $\delta^{13}C$ and $NO_2$, $NO_3^-$ and EC/TC ratio, so we believe that it is not pointless. However, it is possible that these results can be biased by lower content of EC in TC thus we modified part of last sentence at the end of related paragraph as follows: *"This result can be biased by the fact that EC constitutes on average 19% of TC during all seasons. However, it is consistent with positive correlations between $\delta^{13}C$ and gaseous $NO_2$, as well as particulate nitrate, which is also significant in autumn to spring. This result is also supported by the negative correlation between $\delta^{13}C$ and EC/TC ratio (r=-0.51), which is significant only in summer."*

Line 476. Sulfate cannot be emitted, reword.

Response: Changed to "*formation of sulfates*"

Line 507. Dry deposition of fine particulate matter is negligible making this assertion pure speculation. Dry deposition of ammonia can only occur on surfaces - were these frosted surfaces? Confusing interpretation, please reword.

Response: Thank you for this comment. It is right that dry deposition of fine particulate matter is negligible, however, in mentioned part of text we discussed possible dry deposition of gaseous ammonia. Moreover, given sentence is a reference to observations of Savard et al. (2017). Throughout the Event, the temperature was below 0°C (see Fig.9), so frosted surfaces could be possible on the ground surface, which can support deposition but also decrease fluxes of ammonia from e.g. water surfaces or soil. Also the results indicate that during the Event gradually rose deficit of ammonia and after some time the main source of ammonia were probably agricultural emissions from farms whose emissions of $NH_3$ are not as affected by low temperatures. Nevertheless, because we did not measure ammonia fluxes during the campaign, the conclusions on dry deposition heavier isotope of ammonia may sound little bit speculative (as the reviewer mentioned). In this sense we changed text in subsection 3.4 and in Conclusions.

[revised manuscript text omitted]

Siegfried, T.: P value ban: small step for a journal, giant leap for science, Science News, 2015. on-line: https://www.sciencenews.org/blog/context/p-value-ban-small-step-journal-giant-leap-science?mode=blog&context=117

[revised manuscript text omitted]

formation

| Page 9: [2] Deleted | Petr Vodicka | 11/01/2019 14:48:00 |
|---|---|---|

formation

| Page 9: [2] Deleted | Petr Vodicka | 11/01/2019 14:48:00 |
|---|---|---|

formation

| Page 9: [3] Deleted | Petr Vodicka | 11/01/2019 14:48:00 |
|---|---|---|

show

| Page 9: [3] Deleted | Petr Vodicka | 11/01/2019 14:48:00 |
|---|---|---|

show

| Page 9: [3] Deleted | Petr Vodicka | 11/01/2019 14:48:00 |
|---|---|---|

show

| Page 9: [3] Deleted | Petr Vodicka | 11/01/2019 14:48:00 |
|---|---|---|

show

| Page 9: [3] Deleted | Petr Vodicka | 11/01/2019 14:48:00 |
|---|---|---|

show

| Page 9: [3] Deleted | Petr Vodicka | 11/01/2019 14:48:00 |
|---|---|---|

show

| Page 9: [3] Deleted | Petr Vodicka | 11/01/2019 14:48:00 |
|---|---|---|

show

| Page 9: [3] Deleted | Petr Vodicka | 11/01/2019 14:48:00 |
|---|---|---|

show

| Page 9: [3] Deleted | Petr Vodicka | 11/01/2019 14:48:00 |
|---|---|---|

show

| Page 9: [3] Deleted | Petr Vodicka | 11/01/2019 14:48:00 |
|---|---|---|

show

| Page 9: [4] Deleted | Petr Vodicka | 11/01/2019 14:48:00 |
|---|---|---|

generally

| Page 9: [4] Deleted | Petr Vodicka | 11/01/2019 14:48:00 |

generally

| Page 9: [4] Deleted | Petr Vodicka | 11/01/2019 14:48:00 |

generally

| Page 9: [4] Deleted | Petr Vodicka | 11/01/2019 14:48:00 |

generally

| Page 9: [4] Deleted | Petr Vodicka | 11/01/2019 14:48:00 |

generally

| Page 9: [4] Deleted | Petr Vodicka | 11/01/2019 14:48:00 |

generally

| Page 9: [4] Deleted | Petr Vodicka | 11/01/2019 14:48:00 |

generally

| Page 9: [4] Deleted | Petr Vodicka | 11/01/2019 14:48:00 |

generally

| Page 9: [4] Deleted | Petr Vodicka | 11/01/2019 14:48:00 |

generally

| Page 9: [5] Deleted | Petr Vodicka | 11/01/2019 14:48:00 |

in summer,

| Page 9: [5] Deleted | Petr Vodicka | 11/01/2019 14:48:00 |

in summer,

| Page 9: [5] Deleted | Petr Vodicka | 11/01/2019 14:48:00 |

in summer,

| Page 9: [5] Deleted | Petr Vodicka | 11/01/2019 14:48:00 |

in summer,

| Page 9: [5] Deleted | Petr Vodicka | 11/01/2019 14:48:00 |

in summer,

| Page 9: [5] Deleted | Petr Vodicka | 11/01/2019 14:48:00 |

in summer,

| Page 9: [5] Deleted | Petr Vodicka | 11/01/2019 14:48:00 |

in summer,

| Page 9: [5] Deleted | Petr Vodicka | 11/01/2019 14:48:00 |

in summer,

| Page 9: [5] Deleted | Petr Vodicka | 11/01/2019 14:48:00 |

in summer,

| Page 9: [5] Deleted | Petr Vodicka | 11/01/2019 14:48:00 |

in summer,

| Page 9: [5] Deleted | Petr Vodicka | 11/01/2019 14:48:00 |

in summer,

| Page 9: [5] Deleted | Petr Vodicka | 11/01/2019 14:48:00 |

in summer,

| Page 9: [5] Deleted | Petr Vodicka | 11/01/2019 14:48:00 |

in summer,

| Page 9: [5] Deleted | Petr Vodicka | 11/01/2019 14:48:00 |

in summer,

| Page 9: [5] Deleted | Petr Vodicka | 11/01/2019 14:48:00 |

in summer,

| Page 9: [5] Deleted | Petr Vodicka | 11/01/2019 14:48:00 |

in summer,

| Page 9: [5] Deleted | Petr Vodicka | 11/01/2019 14:48:00 |

in summer,

| Page 9: [5] Deleted | Petr Vodicka | 11/01/2019 14:48:00 |

in summer,

| Page 9: [5] Deleted | Petr Vodicka | 11/01/2019 14:48:00 |

in summer,

| Page 9: [5] Deleted | Petr Vodicka | 11/01/2019 14:48:00 |

in summer,

| Page 9: [5] Deleted | Petr Vodicka | 11/01/2019 14:48:00 |

in summer,

| Page 9: [5] Deleted | Petr Vodicka | 11/01/2019 14:48:00 |

in summer,

| Page 9: [5] Deleted | Petr Vodicka | 11/01/2019 14:48:00 |
|---|---|---|

in summer,

---

## Author Comment (AC2) · 13 Jan 2019

Response to anonymous Referee RC2

We would like to thank the reviewer for his valuable and helpful comments. Based on this, we have changed a part of the manuscript which led to an improvement in the final text. Responses to specific comments are below.

This paper presents seasonal variations of d15N and d13C in ambient aerosol collected in Košetice (Central Europe) between 27 September 2013 and 9 August 2014. The authors show an impressive series of measurements aiming to investigate sources and processing of the fine fraction of aerosol at a rural background site. This study using two-isotope analysis is very suitable for this goal. The use of multiple isotope ratios for the study of atmospheric pollution and the chemistry of organic compounds in the atmosphere is a newly emerging tool. The manuscript contributes to scientific progress within the scope of the journal; therefore, it is suitable to be published for discussions in ACP. Both description and discussion of measurements are well founded. Unfortunately, the presentation is not on the same level, therefore it needs to be substantially improved before publishing.

General comments: 1) The authors discuss the benefits of using isotopes in the atmospheric research. These can give some hints to information, which is not available from concentration measurements, such as the impact of sources vs. processing on measured delta values. I miss though a discussion on the current limitations of using isotope ratio measurements for the above mentioned purpose. This omission might be the reason why the interpretation sounds sometimes so futile. Example: Lines262-263 'In the case of our data, mixing of all of these factors probably had an influence on the nitrate isotopic composition during different parts of the year.' Reformulate!

Response: Thank you for this comment. We added following text referring to the current limitations of using isotope ratio measurements in first paragraph of Introduction chapter. "However, studies based on single isotope analysis have their limitations (Meier-Augenstein and Kemp, 2012). Those include an uncertainty when multiple sources or different processes are present, whose measured delta values may overlap (typically in the narrower $\delta 13C$ range). Another factor are isotope fractionation processes which may constrain the accuracy of source identification (Xue et al., 2009). Using isotope analysis on multiple phases (gas and particulate matter) or multiple isotope analysis can overcome these problems and may be useful to constrain the potential sources/processes." Specific text on lines 262-263 were changed to the following sentence: "In our study, it is most likely that all these factors contributed, to a certain extent, to the nitrogen isotopic composition of $NO_3-$ throughout the year."

[Figure]

2) The introduction should make the reader aware of the importance of using multiple isotopes (literature sources are required), e.g. for constraining potential sources. The sentence on the Lines 85-86 is too late and too less. A proper foreword would bring more structure in the discussion from Lines59-83. Here the authors must clearly differentiate between single and multiple isotope analyses.

Response: We extended Introduction chapter about examples of studies using multi isotope analyses. The sentence from the lines 85-86 we slightly modified and moved at the end of this paragraph. New text related to studies with multiple isotope measurements in Introduction chapter is following: "Recently, the multiple isotope approach was applied in several studies by using $\delta$13C and $\delta$15N measurements. Specifically, the $\delta$13C and $\delta$15N composition of aerosol (along with other supporting data) was used to identify the sources and processes on marine sites in Asia (Bikkina et al., 2016; Kunwar et al., 2016; Miyazaki et al., 2011; Xiao et al., 2018). Same isotopes were used to determine the contribution of biomass burning to organic aerosols in India (Boreddy et al., 2018) and in Tanzania (Mkoma et al., 2014), or to unravel the sources of aerosol contamination at Cuban rural and urban coastal sites (Morera-Gómez et al., 2018). These studies show the potential advantages of $\delta$13C and $\delta$15N isotope ratios to characterize aerosol types and to reveal the underlying chemical processes that take place in them." We also added data on other isotope analyzes (if were performed) to distinguish single- and multi-isotope studies in paragraph related to European studies

3) Separate Spearman from Pearson correlation coefficients. For that purpose, label them for each use (e.g. in Line203).

Response: Thank you for this comment. We identified in text few Pearson's correlation coefficients (connected with Figures 2, 3 and 7) instead of Spearman's ones. Although each of these coefficients provides different information (Pearson benchmarks linear relationship, Spearman benchmarks monotonic relationship), the values in our work are same or similar (e.g. for TC vs.TN is r(P): 0.70 and r(S): 0.71). Based on this, we decided to use only Spearman correlation coefficients in this work. Changes were made in Figures 2, 3 and 7, and related text (original lines 203, 213). Currently, Spearman's correlations are used throughout the document so there is no need to differentiate it from Pearson's correlations.

4) Name the described variables throughout the manuscript! Some examples: Line122: Replace 'Determination of TC, TN and their stable isotopes' by 'Determination of TC, TN concentrations and their stable isotope ratios' Line123: Replace 'For the TC and TN analyses' by ' For the TC and TN concentration and isotopic ratio measurements'

Response: For a text clarification, there were changed variables description on following lines: 122, 123, 202, 205, 290, 545,

5) Vague statements should be replaced by precise explanations throughout the paper. An example: Line382: specify the 'secondary processes'

Response: We are sorry for vague statements. We rephrased the text as below. Statement on line 382 was based on work of Widory (2007). We changed the previous sentence to following: "Similarly, the contradictory dependence between $\delta$15N and TN in summer and winter was observed by Widory (2007) in PM10 samples from Paris. Widory (2007) connected this result with different primary nitrogen origin (road-traffic emissions in summer and no specific source in winter) and following secondary processes associated with isotope fractionation during degradation of atmospheric NOx leading to two distinct pathways for 15N enrichment (summer) and depletion (winter)."

R: 6) Generally: swap the negative numbers in ranges. The lower numbers stay first. Examples: Line520 -40 to -28permil and Line522 -38 to -22permil

Response: Ranges of negative numbers were swapped on original lines 338, 520, 522 and 533. Thank you for your notice.

Specific comments: Lines54-57: Reformulate! The OC/EC ratios are very different in aerosol, depending on its sources. Moreover, make more sentences of this single one. Differentiate between equilibrium and kinetic isotopic effect. Guide the reader through that by giving some information on corresponding fractionation (non-equilibrium partitioning causes much lower fractionation than chemical reactions. Contrarily, equilibrium fractionation might be significant).

Response: We changed the Introduction chapter with the text related to isotopes in carbonaceous aerosols and we also inserted a new paragraph on isotope fractionation:

New text related to carbonaceous aerosols: " Total carbon in aerosol is usually divided into elemental carbon (EC) and organic carbon (OC), where OC forms the major part of TC (e.g., Mbengue et al., 2018). Although EC is more or less inert to chemical changes, slightly different $\delta$13C in EC originating from primary emissions are described (Kawashima and Haneishi, 2012). OC represents a wide variety of organic compounds which can originate from different sources with different 13C content resulting in different $\delta$13C values in bulk of emissions. Changes in isotopic ratio of $\delta$13C in OC (and thus also TC) can subsequently affect chemical reactions where isotope fractionations via the kinetic isotope effect (KIE) usually dominate the partitioning between gas and aerosol (liquid/solid) phases (e.g. Zhang et al., 2016)."

New paragraph related to isotope fractionation: "Isotopes are furthermore altered mainly by kinetic and/or equilibrium fractionation processes. Kinetic isotope effects (KIE) occur mainly during unidirectional (irreversible) reactions but also diffusion or during reversible reactions that are not yet at equilibrium (Gensch et al., 2014). Owing to KIE, reaction products (both gasses and particles) are depleted in the heavy isotope relatively to the reactants, and this effect is generally observed in organic compounds (Irei et al., 2006). If the partitioning between phases is caused by non-equilibrium processes (such as e.g. absorption), the isotopic fractionation is small and lower than that caused by chemical reactions (Rahn and Eiler, 2001). Equilibrium isotope effects occur in reversible chemical reactions or phase changes if the system is in equilibrium. Under such conditions, the heavier isotope is bound into the compounds where the total energy of the system is minimized and the most stable. Equilibrium effects are typical for inorganic species and usually temperature dependent."

Line87: No need to introduce TC and TN. It happened already in Lines12-13

Response: Edited and only shortcuts were kept.

Line127: I don't understand. Is the oven temperature 1000 ◦C? How can the marble burn, if that needs 1400 ◦ C?

Response: Theory is that burning tin should locally increase temperature around the sample to approximately $1400°C$, however, this is not so important and it can be also confusing so we deleted temperature $1400°C$ from the MS.

Line131: What does 'parts' means? Give the approximate fraction in %.

Response: At this point, there is an auto-dilution system on the ConFlo IV interface, which is applied to each gas species matching sample and reference gas intensities. This dilution is automatic and the device dynamically reacts to the sample volume. Therefore, it is not possible to specify the exact part of the sample. For this reason, we decided to shorten the sentence to the following form: "Parts of $CO_2$ and $N_2$ were then transferred into an isotope ratio mass spectrometer (IRMS; Delta V, Thermo Fisher Scientific) through a ConFlo IV interface to monitor $15N/14N$ and $13C/12C$ ratios."

Lines135-139: Mention that the final delta values are expressed relatively to the international standards and not to the 'working' standard.

Response: That's a good point. Sentence before equations was extended to following form: 'Subsequently, $\delta15N$ of TN and $\delta13C$ of TC were calculated using the following equations and the final $\delta$ values are expressed in relation to the international standards:'

Line146: The loads on the quartz filter are meant here of course.

Response: Yes, you are right that the loads on quartz filters was analyzed. The sentence was changed in this sense.

Lines198-200: Move these sentences to the first paragraph, they don't belong to Fig.1.

Response: You are right, these sentences belong to Table 1 so we just changed link to Fig.1 to Tab.1 at the end of this paragraph.

Lines218-219: Reformulate: 'but they are still in line with the linear fitting of all annual data'. This is not appropriate.

Response: The sentence was reworded to the following: "The winter Event measurements show the highest $\delta$13C and the lowest $\delta$15N, but a linear fit does not show a significant differences as compared to rest of the data (Fig. 2, right)."

Lines290-291: Reformulate! Either state that the samples containing the highest NO3- concentration show a d15N of..., or fit a histogram plot showing a peak of measurements with NO3- concentrations higher than... at a delta value of 14+/-1 permil.

Response: Thank you for this comment. You are right that statement on lines 290-291 "The $\delta$15N shows a peak at approximately 14±1‰:." is not exact, and is the result of estimation based on exponential curves in Figure 4. So newly, we took samples with NO3- concentrations higher than 6 $\mu$g/m3 (n=5) and we calculated an average $\delta$15N value from these samples. It results in new value of $\delta$15N (13.3±0.7‰, we used this calculated value instead of 14±1‰ in whole text. New text on lines 290-291 is following: "Samples with the highest NO3- concentrations (>6 $\mu$g/m3, n=5) show an average $\delta$15N of 13.3±0.7‰."

Lines300-307: The paragraph should be moved upward to Fig. 3.

Response: The paragraph on lines 300-307 relates to the previous one, where the results in Figure 4 are commented. For this reason, we would like keep this paragraph in the current position.

Lines338-349: Completely rearrange! Suggestion: start with a statement 'The measured TC d13C ranged between.... These values are ... (in which part?) situated in the reported ranges... (here give an overall range. for that take the information from e.g. the review by Gensch et al. 2014). This broad range can be explained by... (plants, marine, combustion sources... whatever). (At this point bring the similarity to other european reported values).'

Response: Thank you for this suggestion. Based on this, paragraph was rearranged as below: "The $\delta$13C of TC ranged from -28.9 to -25.4‰ (Fig. 6) and the lowest $\delta$13C we observed in field blank samples (mean -29.2‰ n=7), indicating that the lowest summer values in particulate matter were close to gas phase values. Our $\delta$13C values are within the range reported for particulate TC (-29‰ to -15‰ as summarized by Gensch et al. (2014). The lowest values are associated with fine particles after combustion and transport (Ancelet et al., 2011; Widory, 2006) while the highest values are associated with the coarse fraction and carbonate contribution (Kawamura et al., 2004). This broad range can be explained by the influence of marine aerosols (Ceburnis et al., 2016), different anthropogenic sources (e.g., Widory et al., 2004), as well as different distributions of C3 and C4 plants (Martinelli et al., 2002) resulting in different $\delta$13C values in the northern and southern hemispheres (Cachier, 1989). The $\delta$13C values at the Košetice site fall within the range common to other European sites. For example, a rural background site in Vavihill (southern Sweden, range -26.7 to -25.6‰ Martinsson et al. (2017)), urban Wroclaw (Poland, range -27.6 to -25.3‰ Górka et al. (2014)), different sites (urban, coastal, forest) in Lithuania (East Europe, Masalaite et al., 2015, 2017), as well as urban Zurich (Switzerland, Fisseha et al. (2009))."

Line349: Replace 'The d13C values are significantly smaller than those of d15N due to' by 'The range of TC d13C values is significantly smaller than that of TN d15N due to'

Response: The sentence was changed based on the comment as bellow: "The range of TC $\delta$13C values is significantly narrower than that of TN $\delta$15N due to..."

Lines358-359: This comparison is confusing: what do you mean? Similar to what? Do you refer the first or the second sentence?

Response: The comparison was between first and third sentence. Second sentence was moved to first paragraph of section 3.2, so now it should be clearer. See first paragraph of subsection 3.2 in the revised MS.

Lines365-370: Change the order of these two sentences. Describe first the observations and then give the explanation.

Response: The order of sentences was changed. See end paragraph of subsection 3.2 in the revised MS.

Line 375: Replace 'these isotopes' with 'isotope distributions'.

Response: Replaced

Lines379-380: Not the changes in aerosol chemistry are different, but the chemistry itself.

Response: Word "changes" was deleted.

Lines386-391: Change the order of the first two sentences. The third one describes the first not the second one.

Response: The order of sentences was changed.

Lines415-422: Lack of clarity! Reformulate, by bringing some structure in it: starting at high NH4/SO4 down to 2 and lower than 2! For each range: particle components, processes (e.g. NH3 deficit in gas phase at ratios <2), seasonal dependence.

Response: Thank you for your suggestion. Based on this comment, we decided to completely change this paragraph to the following: "Figure 8 shows an enrichment of 15N as a function of the molar ratio of NH4+/SO42-. The highest NH4+/SO42- ratios, showing an ammonia rich atmosphere, were observed during winter, late autumn and early spring along with high abundance of NO3- that is related to favorable thermodynamic conditions during heating season and enough ammonia in the atmosphere. Gradual decreasing molar ratios of NH4+/SO42- during spring indicate a gradual increase of ambient temperatures and therefore worsened thermodynamic conditions for NO3- formation in aerosol phase, which was accompanied by a visible decrease in the nitrate content in aerosols (Fig. 8). The increase of temperatures finally leads to the NH4+/SO42- ratio reaching 2 at the turn of spring and summer. Finally, summer values of NH4+/SO42- molar ratio below 2 indicate that SO42- in aerosol particles at high summer temperatures may not be completely saturated with ammonium but it can be composed from mixture of (NH4)2SO4 and NH4HSO4 (Weber et al., 2016). The equilibrium reaction between these two forms of ammonium sulfates related to temperature oscillation during a day and due to vertical mixing of the atmosphere is a probable factor which leads to increased values of $\delta$15N in early summer. Ammonia measurements, that were carried out at the Košetice site until 2001, showed that NH3 concentrations in summer were slightly higher than in winter (http://portal.chmi.cz/files/portal/docs/uoco/isko/tab_roc/2000_enh/CZE/kap_18/kap_18_026.html), which supports temperature as a main factor influencing NH4+/SO42- ratio at Košetice. In this context, we noticed that 25 out of 33 summer samples have molar ratios of NH4+/SO42- below 2, and the remaining samples are approximately 2, and the relative abundance of NO3- in PM1 in those samples is very low (ca. 1.7 %)."

Lines429-434: Too abrupt! Start with the observation of similar gaseous NH3 in summer and winter. Describe what a thermodynamic equilibrium would mean for the particles and how would this be reflected in the delta values. Measurements show a different situation -> more organics in summer...

Response: Thank you for this notice. We moved the sentence, about similar gaseous NH3 concentrations in summer and winter at the Košetice site, to paragraph above (see response to previous comment). The lines have been reformulated into the following form: " In thermodynamic equilibrium, partitioning between gas (NH3) and aerosol (NH4+) phases should result in even larger $\delta$15N values of particles in summer, however, measurements show a different situation. Summer $\delta$15N values are highest but further enrichment was stopped. Moreover, we observed a positive (and significant) correlation between temperature and $\delta$13C (r=0.39) only in summer, whereas the correlation coefficient of $\delta$15N vs. temperature is statistically insignificant, suggesting that while values of $\delta$15N reached their maxima, the $\delta$13C can still grow with even higher temperatures due to the influence of organics in summer season."

Lines482-484: Very confuse sentence. Reformulate!

Response: The sentence is reworded to the following: "During the Event, $\delta$15N correlates positively with NO3- (r=0.96) and NO3–N/TN (0.98). Before the Event, we also observed the highest values of $\delta$15N at approximately 13.3‰ which we previously interpreted by the emissions from domestic heating via coal and/or biomass burning."

Lines570-574: The winter observation should stay before the summer ones. In that way, the flow is more coherent (e.g. no need to explain lower values of TN d15N when there are high fraction of nitrates.).

Response: In summary and conclusions, we wanted to discuss all seasons for the first time. After summary related to seasonal data follow conclusions related to winter Event. This is reason why the winter data are discussed after summer data. It seems more logic to us if the winter Event summary follows winter data than to discuss first about winter, then continue about summer season, and after that return to discuss about the winter Event. So we prefer not to move this paragraph.

Editorial revisions: The used English is not optimal. I do not give any editorial advises! My only suggestion is that this manuscript MUST be carefully revised by a native speaker. The work is too good to risk to make the reader hostile due to the language.

Response: We are sorry for inconvenience with English. In fact, the text was checked by the professional language service before the first submission so we expected it should be alright. As the final step after this review process, we sent again the manuscript for English corrections to Sean Mark Miller who is a native speaker and professional corrector.

The manuscript is 'peppered' with: 1) Wrong prepositions - Lines43-44 'Key processes in the atmosphere, which are involved WITH climate changes, air quality, rain events (Fuzzi et al., 2015) or visibility (Hyslop, 2009), are strongly influenced by aerosols. ' – Lines391-392 ' Although Savard et al. (2017) reported a similar negative d15N in NH4+ dependence AT temperatures in Alberta (Canada),...' Also the word order is wrong.

Response: Text with above mentioned prepositions was changed. Lines43-44: "Aerosols have a strong impact on key processes in the atmosphere associated with climate change..." Lines391-392: "Although Savard et al. (2017) reported a similar negative temperature dependence for $\delta$15N in NH4+ in Alberta (Canada), ..."

2) Unhandy expressions - Lines325-328:' During the domestic heating season with the highest concentrations of NO3and NH4+, we can observe a significant increase in OrgN with $\delta$15N again at approximately 14‰ which implies that the isotopic composition of OrgN is determined by the same process during maximal NO3- concentrations, that is, emissions from domestic heating.'

Response: Sentence on lines325-328 was shortened to: "During the domestic heating season with the highest concentrations of NO3- and NH4+, we can observe a significant increase in OrgN with $\delta$15N again at approximately 13.3‰ which implies that the isotopic composition of OrgN is determined by the same source."

3) Long, confusing sentences Lines361-365 or Lines391-396. In these cases it helps to divide into more clear sentences.

Response: We divided above mentioned long sentences and also others in the revised MS. Newly for line 361-365: "
[revised manuscript text omitted]

Please also note the supplement to this comment:
https://www.atmos-chem-phys-discuss.net/acp-2018-604/acp-2018-604-AC2- supplement.pdf

**Supplement:**

**Response to anonymous Referee RC2**

We would like to thank the reviewer for his valuable and helpful comments. Based on this, we have changed a part of the manuscript which led to an improvement in the final text. Responses to specific comments are below.

This paper presents seasonal variations of d15N and d13C in ambient aerosol collected in Košetice (Central Europe) between 27 September 2013 and 9 August 2014. The authors show an impressive series of measurements aiming to investigate sources and processing of the fine fraction of aerosol at a rural background site. This study using two-isotope analysis is very suitable for this goal.

The use of multiple isotope ratios for the study of atmospheric pollution and the chemistry of organic compounds in the atmosphere is a newly emerging tool. The manuscript contributes to scientific progress within the scope of the journal; therefore, it is suitable to be published for discussions in ACP. Both description and discussion of measurements are well founded. Unfortunately, the presentation is not on the same level, therefore it needs to be substantially improved before publishing.

General comments:

1) The authors discuss the benefits of using isotopes in the atmospheric research. These can give some hints to information, which is not available from concentration measurements, such as the impact of sources vs. processing on measured delta values. I miss though a discussion on the current limitations of using isotope ratio measurements for the above mentioned purpose. This omission might be the reason why the interpretation sounds sometimes so futile.

Example: Lines262-263 'In the case of our data, mixing of all of these factors probably had an influence on the nitrate isotopic composition during different parts of the year.'

Reformulate!

Response: Thank you for this comment. We added following text referring to the current limitations of using isotope ratio measurements in first paragraph of Introduction chapter.

*"However, studies based on single isotope analysis have their limitations (Meier-Augenstein and Kemp, 2012). Those include an uncertainty when multiple sources or different processes are present, whose measured delta values may overlap (typically in the narrower $\delta^{13}C$ range). Another factor are isotope fractionation processes which may constrain the accuracy of source identification (Xue et al., 2009). Using isotope analysis on multiple phases (gas and particulate matter) or multiple isotope analysis can overcome these problems and may be useful to constrain the potential sources/processes."*

Specific text on lines 262-263 were changed to the following sentence: *"In our study, it is most likely that all these factors contributed, to a certain extent, to the nitrogen isotopic composition of $NO_3^-$ throughout the year."*

2) The introduction should make the reader aware of the importance of using multiple isotopes (literature sources are required), e.g. for constraining potential sources. The sentence on the Lines 85- is too late and too less. A proper foreword would bring more structure in the discussion from Lines59-83. Here the authors must clearly differentiate between single and multiple isotope analyses.

Response: We extended Introduction chapter about examples of studies using multi isotope analyses. The sentence from the lines 85-86 we slightly modified and moved at the end of this paragraph. New text related to studies with multiple isotope measurements in Introduction chapter is following:

"Recently, the multiple isotope approach was applied in several studies by using $\delta^{13}C$ and $\delta^{15}N$ measurements. Specifically, the $\delta^{13}C$ and $\delta^{15}N$ composition of aerosol (along with other supporting data) was used to identify the sources and processes on marine sites in Asia (Bikkina et al., 2016; Kunwar et al., 2016; Miyazaki et al., 2011; Xiao et al., 2018). Same isotopes were used to determine the contribution of biomass burning to organic aerosols in India (Boreddy et al., 2018) and in Tanzania (Mkoma et al., 2014), or to unravel the sources of aerosol contamination at Cuban rural and urban coastal sites (Morera-Gómez et al., 2018). These studies show the potential advantages of $\delta^{13}C$ and $\delta^{15}N$ isotope ratios to characterize aerosol types and to reveal the underlying chemical processes that take place in them."

We also added data on other isotope analyzes (if were performed) to distinguish single- and multi-isotope studies in paragraph related to European studies

3) Separate Spearman from Pearson correlation coefficients. For that purpose, label them for each use (e.g. in Line203).

Response: Thank you for this comment. We identified in text few Pearson's correlation coefficients (connected with Figures 2, 3 and 7) instead of Spearman's ones. Although each of these coefficients provides different information (Pearson benchmarks linear relationship, Spearman benchmarks monotonic relationship), the values in our work are same or similar (e.g. for TC vs.TN is r(P): 0.70 and r(S): 0.71). Based on this, we decided to use only Spearman correlation coefficients in this work. Changes were made in Figures 2, 3 and 7, and related text (original lines 203, 213). Currently, Spearman's correlations are used throughout the document so there is no need to differentiate it from Pearson's correlations.

4) Name the described variables throughout the manuscript!

Some examples: Line122: Replace 'Determination of TC, TN and their stable isotopes'

by 'Determination of TC, TN concentrations and their stable isotope ratios'

Line123: Replace 'For the TC and TN analyses' by ' For the TC and TN concentration and isotopic ratio measurements'

Response: For a text clarification, there were changed variables description on following lines: 122, 123, 202, 205, 290, 545,

5) Vague statements should be replaced by precise explanations throughout the paper.

An example: Line382: specify the 'secondary processes'

Response: We are sorry for vague statements. We rephrased the text as below.

Statement on line 382 was based on work of Widory (2007). We changed the previous sentence to following: *"Similarly, the contradictory dependence between δ¹⁵N and TN in summer and winter was observed by Widory (2007) in PM10 samples from Paris. Widory (2007) connected this result with different primary nitrogen origin (road-traffic emissions in summer and no specific source in winter) and following secondary processes associated with isotope fractionation during degradation of atmospheric NOx leading to two distinct pathways for ¹⁵N enrichment (summer) and depletion (winter)."*

R: 6) Generally: swap the negative numbers in ranges. The lower numbers stay first.

Examples: Line520 -40 to -28permil and Line522 -38 to -22permil

Response: Ranges of negative numbers were swapped on original lines 338, 520, 522 and 533. Thank you for your notice.

Specific comments:

Lines54-57: Reformulate! The OC/EC ratios are very different in aerosol, depending on its sources. Moreover, make more sentences of this single one. Differentiate between equilibrium and kinetic isotopic effect. Guide the reader through that by giving some information on corresponding fractionation (non-equilibrium partitioning causes much lower fractionation than chemical reactions. Contrarily, equilibrium fractionation might be significant).

Response: We changed the Introduction chapter with the text related to isotopes in carbonaceous aerosols and we also inserted a new paragraph on isotope fractionation:

New text related to carbonaceous aerosols:
*" Total carbon in aerosol is usually divided into elemental carbon (EC) and organic carbon (OC), where OC forms the major part of TC (e.g., Mbengue et al., 2018). Although EC is more or less inert to chemical changes, slightly different δ¹³C in EC originating from primary emissions are described (Kawashima and Haneishi, 2012). OC represents a wide variety of organic compounds which can originate from different sources with different ¹³C content resulting in different δ¹³C values in bulk of emissions. Changes in isotopic ratio of δ¹³C in OC (and thus also TC) can subsequently affect chemical reactions where isotope fractionations via the kinetic isotope effect (KIE) usually dominate the partitioning between gas and aerosol (liquid/solid) phases (e.g. Zhang et al., 2016)."*

New paragraph related to isotope fractionation:
*"Isotopes are furthermore altered mainly by kinetic and/or equilibrium fractionation processes. Kinetic isotope effects (KIE) occur mainly during unidirectional (irreversible) reactions but also diffusion or during reversible reactions that are not yet at equilibrium (Gensch et al., 2014). Owing to KIE, reaction products (both gasses and particles) are depleted in the heavy isotope relatively to the reactants, and this effect is generally observed in organic compounds (Irei et al., 2006). If the partitioning between phases is caused by non-equilibrium processes (such as e.g. absorption), the isotopic fractionation is small and lower than that caused by chemical reactions (Rahn and Eiler, 2001). Equilibrium isotope effects occur in reversible chemical reactions or phase changes if the system is in equilibrium. Under such conditions, the heavier isotope is bound into the compounds where the total energy of the system*

*is minimized and the most stable. Equilibrium effects are typical for inorganic species and usually temperature dependent."*

Line87: No need to introduce TC and TN. It happened already in Lines12-13

Response: Edited and only shortcuts were kept.

Line127: I don't understand. Is the oven temperature 1000 ∘C? How can the marble burn, if that needs 1400 ∘ C?

Response: Theory is that burning tin should locally increase temperature around the sample to approximately 1400°C, however, this is not so important and it can be also confusing so we deleted temperature 1400°C from the MS.

Line131: What does 'parts' means? Give the approximate fraction in %.

Response: At this point, there is an auto-dilution system on the ConFlo IV interface, which is applied to each gas species matching sample and reference gas intensities. This dilution is automatic and the device dynamically reacts to the sample volume. Therefore, it is not possible to specify the exact part of the sample. For this reason, we decided to shorten the sentence to the following form:

*" $CO_2$ and $N_2$ were then transferred into an isotope ratio mass spectrometer (IRMS; Delta V, Thermo Fisher Scientific) through a ConFlo IV interface to monitor $^{15}N/^{14}N$ and $^{13}C/^{12}C$ ratios."*

Lines135-139: Mention that the final delta values are expressed relatively to the international standards and not to the 'working' standard.

Response: That's a good point. Sentence before equations was extended to following form:

'*Subsequently, $\delta^{15}N$ of TN and $\delta^{13}C$ of TC were calculated using the following equations and the final $\delta$ values are expressed in relation to the international standards:*'

Line146: The loads on the quartz filter are meant here of course.

Response: Yes, you are right that the loads on quartz filters was analyzed. The sentence was changed in this sense.

Lines198-200: Move these sentences to the first paragraph, they don't belong to Fig.1.

Response: You are right, these sentences belong to Table 1 so we just changed link to Fig.1 to Tab.1 at the end of this paragraph.

Lines218-219: Reformulate: 'but they are still in line with the linear fitting of all annual data'. This is not appropriate.

Response: The sentence was reworded to the following: *"The winter Event measurements show the highest $\delta^{13}C$ and the lowest $\delta^{15}N$, but a linear fit does not show a significant differences as compared to rest of the data (Fig. 2, right)."*

Lines290-291: Reformulate! Either state that the samples containing the highest NO3- concentration show a d15N of..., or fit a histogram plot showing a peak of measurements with NO3- concentrations higher than... at a delta value of 14+/-1 permil.

Response: Thank you for this comment. You are right that statement on lines 290-291 *"The $\delta^{15}N$ shows a peak at approximately 14±1‰..."* is not exact, and is the result of estimation based on exponential curves in Figure 4. So newly, we took samples with $NO_3^-$ concentrations higher than 6 μg/m$^3$ (n=5) and we calculated an average $\delta^{15}N$ value from these samples. It results in new value of $\delta^{15}N$ (13.3±0.7‰), we used this calculated value instead of 14±1‰ in whole text.

New text on lines 290-291 is following: *"Samples with the highest $NO_3^-$ concentrations (>6 μg/m$^3$, n=5) show an average $\delta^{15}N$ of 13.3±0.7‰."*

Lines300-307: The paragraph should be moved upward to Fig. 3.

Response: The paragraph on lines 300-307 relates to the previous one, where the results in Figure 4 are commented. For this reason, we would like keep this paragraph in the current position.

Lines338-349: Completely rearrange! Suggestion: start with a statement 'The measured TC d13C ranged between.... These values are ... (in which part?) situated in the reported ranges... (here give an overall range. for that take the information from e.g. the review by Gensch et al. 2014). This broad range can be explained by... (plants, marine, combustion sources... whatever). (At this point bring the similarity to other european reported values).'

Response: Thank you for this suggestion. Based on this, paragraph was rearranged as below:

*"The $\delta^{13}C$ of TC ranged from -28.9 to -25.4‰ (Fig. 6) and the lowest $\delta^{13}C$ we observed in field blank samples (mean -29.2‰, n=7), indicating that the lowest summer values in particulate matter were close to gas phase values. Our $\delta^{13}C$ values are within the range reported for particulate TC (-29‰ to -15‰) as summarized by Gensch et al. (2014). The lowest values are associated with fine particles after combustion and transport (Ancelet et al., 2011; Widory, 2006) while the highest values are associated with the coarse fraction and carbonate contribution (Kawamura et al., 2004). This broad range can be explained by the influence of marine aerosols (Ceburnis et al., 2016), different anthropogenic sources (e.g., Widory et al., 2004), as well as different distributions of C3 and C4 plants (Martinelli et al., 2002) resulting in different $\delta^{13}C$ values in the northern and southern hemispheres (Cachier, 1989). The $\delta^{13}C$ values at the Košetice site fall within the range common to other European sites. For example, a rural background site in Vavihill (southern Sweden, range -26.7 to -25.6‰, Martinsson et al. (2017)), urban Wroclaw (Poland, range -27.6 to -25.3‰, Górka et al. (2014)), different sites (urban, coastal, forest) in Lithuania (East Europe, Masalaite et al., 2015, 2017), as well as urban Zurich (Switzerland, Fisseha et al. (2009))."*

Line349: Replace 'The d13C values are significantly smaller than those of d15N due to' by 'The range of TC d13C values is significantly smaller than that of TN d15N due to'

*Response: The sentence was changed based on the comment as bellow: "The range of TC $\delta^{13}C$ values is significantly narrower than that of TN $\delta^{15}N$ due to..."*

Lines358-359: This comparison is confusing: what do you mean? Similar to what? Do you refer the first or the second sentence?

Response: The comparison was between first and third sentence. Second sentence was moved to first paragraph of section 3.2, so now it should be clearer. See first paragraph of subsection 3.2 in the revised MS.

Lines365-370: Change the order of these two sentences. Describe first the observations and then give the explanation.

Response: The order of sentences was changed. See end paragraph of subsection 3.2 in the revised MS.

Line 375: Replace 'these isotopes' with 'isotope distributions'.

Response: Replaced

Lines379-380: Not the changes in aerosol chemistry are different, but the chemistry itself.

Response: Word "changes" was deleted.

Lines386-391: Change the order of the first two sentences. The third one describes the first not the second one.

Response: The order of sentences was changed.

Lines415-422: Lack of clarity! Reformulate, by bringing some structure in it: starting at high NH4/SO4 down to 2 and lower than 2! For each range: particle components, processes (e.g. NH3 deficit in gas phase at ratios <2), seasonal dependence.

Response: Thank you for your suggestion. Based on this comment, we decided to completely change this paragraph to the following:

*"Figure 8 shows an enrichment of $^{15}N$ as a function of the molar ratio of $NH_4^+/SO_4^{2-}$. The highest $NH_4^+/SO_4^{2-}$ ratios, showing an ammonia rich atmosphere, were observed during winter, late autumn and early spring along with high abundance of $NO_3^-$ that is related to favorable thermodynamic conditions during heating season and enough ammonia in the atmosphere. Gradual decreasing molar ratios of $NH_4^+/SO_4^{2-}$ during spring indicate a gradual increase of ambient temperatures and therefore worsened thermodynamic conditions for $NO_3^-$ formation in aerosol phase, which was accompanied by*

*a visible decrease in the nitrate content in aerosols (Fig. 8). The increase of temperatures finally leads to the $NH_4^+/SO_4^{2-}$ ratio reaching 2 at the turn of spring and summer. Finally, summer values of $NH_4^+/SO_4^{2-}$ molar ratio below 2 indicate that $SO_4^{2-}$ in aerosol particles at high summer temperatures may not be completely saturated with ammonium but it can be composed from mixture of $(NH_4)_2SO_4$ and $NH_4HSO_4$ (Weber et al., 2016). The equilibrium reaction between these two forms of ammonium sulfates related to temperature oscillation during a day and due to vertical mixing of the atmosphere is a probable factor which leads to increased values of $\delta^{15}N$ in early summer. Ammonia measurements, that were carried out at the Košetice site until 2001, showed that $NH_3$ concentrations in summer were slightly higher than in winter (http://portal.chmi.cz/files/portal/docs/uoco/isko/tab_roc/2000_enh/CZE/kap_18/kap_18_026.html ), which supports temperature as a main factor influencing $NH_4^+/SO_4^{2-}$ ratio at Košetice. In this context, we noticed that 25 out of 33 summer samples have molar ratios of $NH_4^+/SO_4^{2-}$ below 2, and the remaining samples are approximately 2, and the relative abundance of $NO_3^-$ in PM1 in those samples is very low (ca. 1.7 %)."*

Lines429-434: Too abrupt! Start with the observation of similar gaseous NH3 in summer and winter. Describe what a thermodynamic equilibrium would mean for the particles and how would this be reflected in the delta values. Measurements show a different situation -> more organics in summer...

Response: Thank you for this notice. We moved the sentence, about similar gaseous $NH_3$ concentrations in summer and winter at the Košetice site, to paragraph above (see response to previous comment). The lines have been reformulated into the following form:

„ *In thermodynamic equilibrium, partitioning between gas ($NH_3$) and aerosol ($NH_4^+$) phases should result in even larger $\delta^{15}N$ values of particles in summer, however, measurements show a different situation. Summer $\delta^{15}N$ values are highest but further enrichment was stopped. Moreover, we observed a positive (and significant) correlation between temperature and $\delta^{13}C$ (r=0.39) only in summer, whereas the correlation coefficient of $\delta^{15}N$ vs. temperature is statistically insignificant, suggesting that while values of $\delta^{15}N$ reached their maxima, the $\delta^{13}C$ can still grow with even higher temperatures due to the influence of organics in summer season."*

Lines482-484: Very confuse sentence. Reformulate!

Response: The sentence is reworded to the following: *"During the Event, $\delta^{15}N$ correlates positively with $NO_3^-$ (r=0.96) and $NO_3^-$-N/TN (0.98). Before the Event, we also observed the highest values of $\delta^{15}N$ at approximately 13.3‰, which we previously interpreted by the emissions from domestic heating via coal and/or biomass burning."*

Lines570-574: The winter observation should stay before the summer ones. In that way, the flow is more coherent (e.g. no need to explain lower values of TN d15N when there are high fraction of nitrates.).

Response: In summary and conclusions, we wanted to discuss all seasons for the first time. After summary related to seasonal data follow conclusions related to winter Event. This is reason why the winter data are discussed after summer data. It seems more logic to us if the winter Event summary follows winter data than to discuss first about winter, then continue about summer season, and after that return to discuss about the winter Event.  So we prefer not to move this paragraph.

Editorial revisions:

The used English is not optimal. I do not give any editorial advises! My only suggestion is that this manuscript MUST be carefully revised by a native speaker. The work is too good to risk to make the reader hostile due to the language.

Response: We are sorry for inconvenience with English. In fact, the text was checked by the professional language service before the first submission so we expected it should be alright. As the final step after this review process, we sent again the manuscript for English corrections to Sean Mark Miller who is a native speaker and professional corrector.

The manuscript is 'peppered' with:

1) Wrong prepositions

- Lines43-44 'Key processes in the atmosphere, which are involved WITH climate changes, air quality, rain events (Fuzzi et al., 2015) or visibility (Hyslop, 2009), are strongly influenced by aerosols. ' –

Lines391-392 ' Although Savard et al. (2017) reported a similar negative d15N in NH4+ dependence AT temperatures in Alberta (Canada),...' Also the word order is wrong.

Response: Text with above mentioned prepositions was changed.

Lines43-44: *"Aerosols have a strong impact on key processes in the atmosphere associated with climate change..."*

Lines391-392: *"Although Savard et al. (2017) reported a similar negative temperature dependence for $\delta^{15}N$ in $NH_4^+$ in Alberta (Canada), ..."*

2) Unhandy expressions

- Lines325-328:' During the domestic heating season with the highest concentrations of NO3and NH4+, we can observe a significant increase in OrgN with δ15N again at approximately 14‰ which implies that the isotopic composition of OrgN is determined by the same process during maximal NO3-concentrations, that is, emissions from domestic heating.'

Response: Sentence on lines325-328 was shortened to: *"During the domestic heating season with the highest concentrations of $NO_3^-$ and $NH_4^+$, we can observe a significant increase in OrgN with $\delta^{15}N$ again at approximately 13.3‰, which implies that the isotopic composition of OrgN is determined by the same source."*

3) Long, confusing sentences

Lines361-365 or Lines391-396. In these cases it helps to divide into more clear sentences.

Response: We divided above mentioned long sentences and also others in the revised MS.

[revised manuscript text omitted]

formation

| **Page 9: [2] Deleted** | **Petr Vodicka** | **11/01/2019 14:48:00** |

formation

| **Page 9: [2] Deleted** | **Petr Vodicka** | **11/01/2019 14:48:00** |

formation

| **Page 9: [3] Deleted** | **Petr Vodicka** | **11/01/2019 14:48:00** |

show

| **Page 9: [3] Deleted** | **Petr Vodicka** | **11/01/2019 14:48:00** |

show

| **Page 9: [3] Deleted** | **Petr Vodicka** | **11/01/2019 14:48:00** |

show

| **Page 9: [3] Deleted** | **Petr Vodicka** | **11/01/2019 14:48:00** |

show

| **Page 9: [3] Deleted** | **Petr Vodicka** | **11/01/2019 14:48:00** |

show

| **Page 9: [3] Deleted** | **Petr Vodicka** | **11/01/2019 14:48:00** |

show

| **Page 9: [3] Deleted** | **Petr Vodicka** | **11/01/2019 14:48:00** |

show

| **Page 9: [3] Deleted** | **Petr Vodicka** | **11/01/2019 14:48:00** |

show

| **Page 9: [3] Deleted** | **Petr Vodicka** | **11/01/2019 14:48:00** |

show

| **Page 9: [3] Deleted** | **Petr Vodicka** | **11/01/2019 14:48:00** |

show

| **Page 9: [4] Deleted** | **Petr Vodicka** | **11/01/2019 14:48:00** |

generally

| Page 9: [4] Deleted | Petr Vodicka | 11/01/2019 14:48:00 |

generally

| Page 9: [4] Deleted | Petr Vodicka | 11/01/2019 14:48:00 |

generally

| Page 9: [4] Deleted | Petr Vodicka | 11/01/2019 14:48:00 |

generally

| Page 9: [4] Deleted | Petr Vodicka | 11/01/2019 14:48:00 |

generally

| Page 9: [4] Deleted | Petr Vodicka | 11/01/2019 14:48:00 |

generally

| Page 9: [4] Deleted | Petr Vodicka | 11/01/2019 14:48:00 |

generally

| Page 9: [4] Deleted | Petr Vodicka | 11/01/2019 14:48:00 |

generally

| Page 9: [4] Deleted | Petr Vodicka | 11/01/2019 14:48:00 |

generally

| Page 9: [5] Deleted | Petr Vodicka | 11/01/2019 14:48:00 |

in summer,

| Page 9: [5] Deleted | Petr Vodicka | 11/01/2019 14:48:00 |

in summer,

| Page 9: [5] Deleted | Petr Vodicka | 11/01/2019 14:48:00 |

in summer,

| Page 9: [5] Deleted | Petr Vodicka | 11/01/2019 14:48:00 |

in summer,

| Page 9: [5] Deleted | Petr Vodicka | 11/01/2019 14:48:00 |

in summer,

| Page 9: [5] Deleted | Petr Vodicka | 11/01/2019 14:48:00 |

in summer,

| Page 9: [5] Deleted | Petr Vodicka | 11/01/2019 14:48:00 |

in summer,

| Page 9: [5] Deleted | Petr Vodicka | 11/01/2019 14:48:00 |

in summer,

| Page 9: [5] Deleted | Petr Vodicka | 11/01/2019 14:48:00 |

in summer,

| Page 9: [5] Deleted | Petr Vodicka | 11/01/2019 14:48:00 |

in summer,

| Page 9: [5] Deleted | Petr Vodicka | 11/01/2019 14:48:00 |

in summer,

| Page 9: [5] Deleted | Petr Vodicka | 11/01/2019 14:48:00 |

in summer,

| Page 9: [5] Deleted | Petr Vodicka | 11/01/2019 14:48:00 |

in summer,

| Page 9: [5] Deleted | Petr Vodicka | 11/01/2019 14:48:00 |

in summer,

| Page 9: [5] Deleted | Petr Vodicka | 11/01/2019 14:48:00 |

in summer,

| Page 9: [5] Deleted | Petr Vodicka | 11/01/2019 14:48:00 |

in summer,

| Page 9: [5] Deleted | Petr Vodicka | 11/01/2019 14:48:00 |

in summer,

| Page 9: [5] Deleted | Petr Vodicka | 11/01/2019 14:48:00 |

in summer,

| Page 9: [5] Deleted | Petr Vodicka | 11/01/2019 14:48:00 |

in summer,

| Page 9: [5] Deleted | Petr Vodicka | 11/01/2019 14:48:00 |

in summer,

| Page 9: [5] Deleted | Petr Vodicka | 11/01/2019 14:48:00 |

in summer,

| Page 9: [5] Deleted | Petr Vodicka | 11/01/2019 14:48:00 |

in summer,

| Page 9: [5] Deleted | Petr Vodicka | 11/01/2019 14:48:00 |
|---|---|---|

in summer,